# Stimulation of hypothalamic oxytocin neurons suppresses colorectal cancer progression in mice

Susu Pan[1,2], Kaili Yin[1,2], Zhiwei Tang[3], Shuren Wang[4], Zhuo Chen[1,2], Yirong Wang[3], Hongxia Zhu[4], Yunyun Han[2,5], Mei Liu[4]*, Man Jiang[3]*, Ningzhi Xu[4]*, Guo Zhang[1,2]*

[1]Key Laboratory of Environmental Health, Ministry of Education, Department of Toxicology, School of Public Health, Tongji Medical College, Wuhan, China; [2]Institute for Brain Research, Collaborative Innovation Center for Brain Science, Huazhong University of Science and Technology, Wuhan, China; [3]Department of Physiology, School of Basic Medicine, Tongji Medical College, Huazhong University of Science and Technology, Wuhan, China; [4]Laboratory of Cell and Molecular Biology, State Key Laboratory of Molecular Oncology, National Cancer Center/National Clinical Research Center for Cancer/Cancer Hospital, Chinese Academy of Medical Sciences and Peking Union Medical College, Beijing, China; [5]Department of Neurobiology, School of Basic Medicine, Tongji Medical College, Huazhong University of Science and Technology, Wuhan, China

*For correspondence:
liumei@cicams.ac.cn (ML);
manjiang@hust.edu.cn (MJ);
xuningzhi@cicams.ac.cn (NX);
gzhang@hust.edu.cn (GZ)

**Abstract** Emerging evidence suggests that the nervous system is involved in tumor development in the periphery, however, the role of the central nervous system remains largely unknown. Here, by combining genetic, chemogenetic, pharmacological, and electrophysiological approaches, we show that hypothalamic oxytocin (Oxt)-producing neurons modulate colitis-associated cancer (CAC) progression in mice. Depletion or activation of Oxt neurons could augment or suppress CAC progression. Importantly, brain treatment with celastrol, a pentacyclic triterpenoid, excites Oxt neurons and inhibits CAC progression, and this anti-tumor effect was significantly attenuated in Oxt neuron-lesioned mice. Furthermore, brain treatment with celastrol suppresses sympathetic neuronal activity in the celiac-superior mesenteric ganglion (CG-SMG), and activation of β2 adrenergic receptor abolishes the anti-tumor effect of Oxt neuron activation or centrally administered celastrol. Taken together, these findings demonstrate that hypothalamic Oxt neurons regulate CAC progression by modulating the neuronal activity in the CG-SMG. Stimulation of Oxt neurons using chemicals, for example, celastrol, might be a novel strategy for colorectal cancer treatment.

## Introduction

Colorectal cancer (CRC) is the third most commonly diagnosed malignant tumor and the second leading cause of cancer death globally. There were 1.8 million new cases, and 900,000 patients died of CRC annually worldwide (*Bray et al., 2018*). It is estimated that there were more than 1.5 million people living with a previous CRC diagnosis in the United States in 2019 (*Miller et al., 2019*), and approximately 147,950 new cases will be diagnosed and 53,200 individuals will die of CRC in 2020 (*Siegel et al., 2020*). Besides, prevalence of CRC is rapidly rising in developing countries. For instance, incidence and mortality of CRC rank third and fifth in both men and women among all cancers in China (*Cao et al., 2020*). Thus, it is imperative to understand the mechanism(s) of CRC development. Negative moods, including anxiety, stress, and depression, are frequently associated with the occurrences of cancers (*Antoni et al., 2006*; *Lillberg et al., 2003*). Anxiety is linked to a greater damage

**eLife digest** Colorectal (or 'bowel') cancer killed nearly a million people in 2018 alone: it is, in fact, the second leading cause of cancer death globally. Lifestyle factors and inflammatory bowel conditions such as chronic colitis can heighten the risk of developing the disease. However, research has also linked to the development of colorectal tumours to stress, anxiety and depression. This 'brain-gut' connection is particularly less-well understood.

One brain region of interest is the hypothalamus, an almond-sized area which helps to regulate mood and bodily processes using chemical messengers that act on various cells in the body. For instance, Oxt neurons in the hypothalamus produce the hormone oxytocin which regulates emotional and social behaviours. These cells play an important role in modulating anxiety, stress and depression.

To investigate whether they could also influence the growth of colorectal tumours, Pan et al. used various approaches to manipulate the activity of Oxt neurons in mice with colitis-associated cancer. Disrupting the Oxt neurons in these animals increased anxiety-like behaviours and promoted tumour growth. Stimulating these cells, on the other hand, suppressed cancer progression.

Further experiments also showed that treating the mice with celastrol, a plant extract which can act on the hypothalamus, stimulated Oxt neurons and reduced tumour growth. In particular, the compound worked by acting on a nerve structure in the abdomen which relays messages to the gut.

These preliminary findings suggest that the hypothalamus and its Oxt-producing neurons may influence the progression of colorectal cancer in mice by regulating the activity of an abdominal 'hub' of the nervous system. Modulating the activity of Oxt-producing neurons could therefore be a potential avenue for treatment.

of adaptive immunity (*Lutgendorf et al., 2008*) and impaired quality of life among cancer patients (*Delgado-Guay et al., 2009*). Stress is related to the incidence or mortality of CRC in women (*Kikuchi et al., 2017*; *Kojima et al., 2005*; *Nielsen et al., 2008*). Although negative mood is associated with the development of cancer, the underlying neural mechanism remains poorly understood.

The hypothalamus is a key brain region in mood regulation (*Price and Drevets, 2010*; *Schindler et al., 2012*). Oxytocin (Oxt) neuropeptide-producing neurons in the paraventricular nucleus (PVN) of the hypothalamus are critical in the regulation of anxiety, stress, and depression (*Neumann, 2008*; *Neumann and Landgraf, 2012*). Previous work demonstrated that Oxt was anxiolytic when administered to humans (*Heinrichs et al., 2003*) and rodents (*Blume et al., 2008*; *Ring et al., 2006*; *Windle et al., 1997*), whereas disruption of *Oxt* gene elevated anxiety level in mice (*Amico et al., 2004*; *Mantella et al., 2003*). Hence, Oxt plays a crucial role in mood control. Recent work indicated that nerve fibers of the autonomous nervous system are critically involved in the progressions of prostate (*Magnon et al., 2013*), stomach (*Hayakawa et al., 2017*), and breast cancers (*Kamiya et al., 2019*). Furthermore, the central nervous system (CNS), in particular the hypothalamus, was shown to regulate peripheral tumor progression (*Cao et al., 2010*). However, the neuronal population(s) involved in this process remain unclear. In this work, by combining genetic, chemogenetic, pharmacological, and electrophysiological approaches, we show that Oxt neurons in the PVN regulate tumor progression in a CRC mouse model.

## Results

### Depletion of Oxt neurons promotes CAC progression

Dysregulation of mood is frequently associated with the occurrences of cancer (*Antoni et al., 2006*; *Lillberg et al., 2003*), while Oxt produced in the hypothalamus has an anxiolytic effect (*Neumann, 2008*; *Neumann and Landgraf, 2012*), suggesting that modulation of Oxt neurons may impact tumor progression in the periphery. To address this possibility, we crossed the *Oxt*$^{Cre}$ (*Wu et al., 2012*) with the *Rosa26*$^{DTA176}$ knockin (*Wu et al., 2006*) mice (*Figure 1A*). By doing so, we obtained *Oxt*$^{Cre}$ and the littermate *Oxt*$^{Cre}$;*Rosa26*$^{DTA176}$ (*Oxt*$^{Cre}$;*DTA*) mice, in which the Oxt-producing neurons in the brain had been depleted (*Figure 1B and C*). To confirm the importance of Oxt neurons in anxiety modulation, we analyzed the anxiety-like behavior of *Oxt*$^{Cre}$ and *Oxt*$^{Cre}$;*DTA* mice. In the open field test, *Oxt*$^{Cre}$;*DTA* mice spent less time in the central region than that of the *Oxt*$^{Cre}$ mice (*Figure 1—figure supplement*

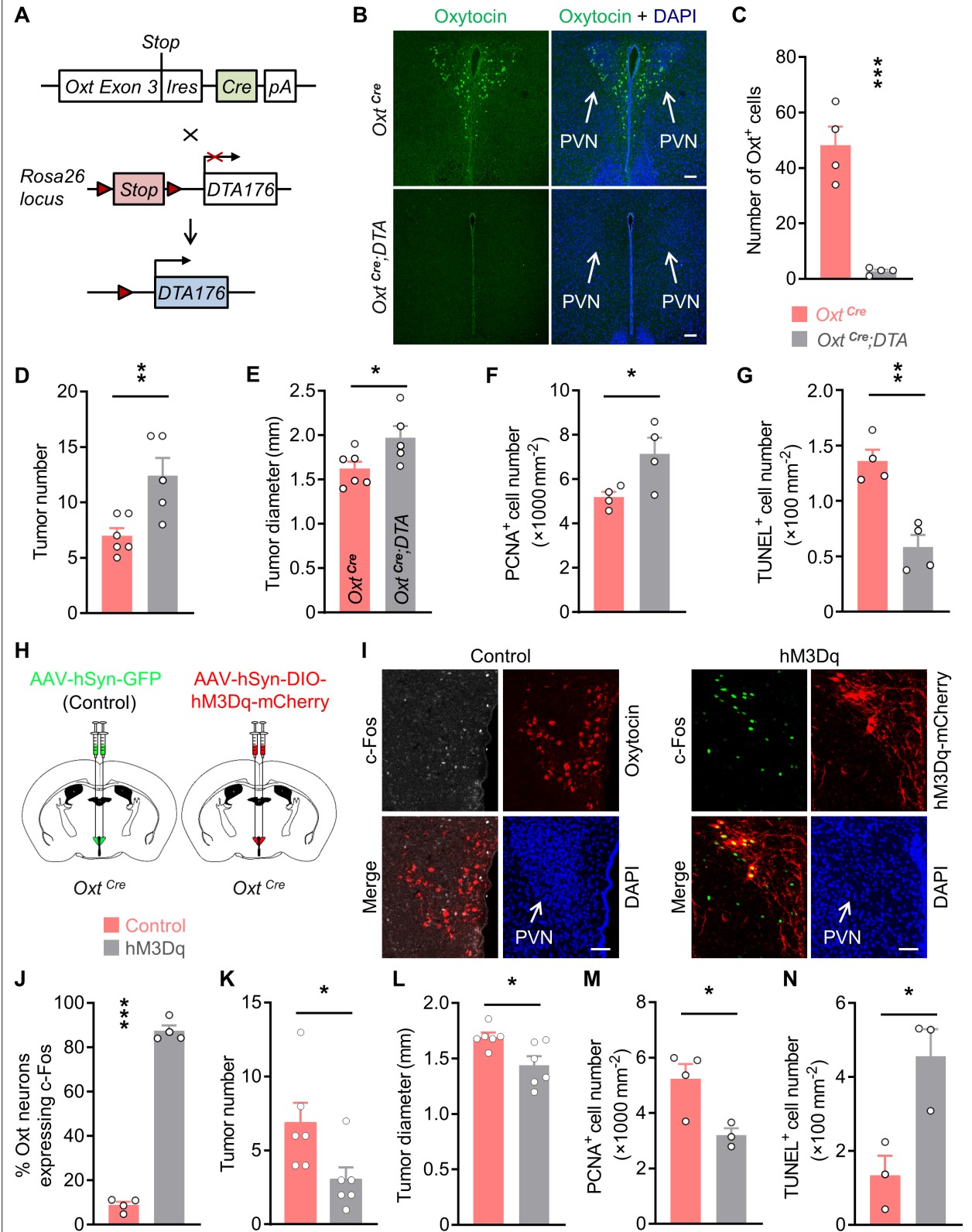

**Figure 1.** Oxytocin (Oxt) neurons modulate the progression of azoxymethane (AOM)/dextran sodium sulfate (DSS)-induced colitis-associated cancer (CAC) in mice. (**A**) A schematic diagram showing the strategy of generating *Oxt^Cre;DTA* mice. When Cre recombinase is present, *loxP*-flanked Stop cassette is excised, therefore allowing the expression of DTA176 in Oxt neurons. Triangles represent *loxP* sites. *Ires*, internal ribosome entry site. *pA*, simian virus 40 polyadenylation signal. (**B**) The CAC was induced in the 2 -month-old *Oxt^Cre* and *Oxt^Cre;DTA* mice using AOM and DSS (see also

*Figure 1 continued*

*Figure 1—figure supplement 1D*). After completing the experiment, immunofluorescent staining for Oxt (green) indicated that Oxt neurons had been depleted in the paraventricular nucleus (PVN) of *Oxt^Cre;DTA* mice. Cell nuclei were counterstained with DAPI (blue). Scale bars, 100 μm. (**C**) The number of Oxt-positive cells in the PVN. n = 4 mice per group. (**D and E**) The CAC was induced in the 2- month-old *Oxt^Cre* and *Oxt^Cre;DTA* mice using AOM and DSS. Tumor number (**D**) and diameter (**E**) in mice treated with AOM/DSS are shown. n = 6 (*Oxt^Cre*) or 5 (*Oxt^Cre;DTA*) mice per group. (**F**) The density of proliferating cell nuclear antigen (PCNA)-positive cells in the tumor tissues of AOM/DSS-treated *Oxt^Cre* and *Oxt^Cre;DTA* mice. n = 4 mice per group. (**G**) The density of terminal deoxynucleotidyl transferase dUTP nick end labeling (TUNEL)-positive cells in tumor tissues. n = 4 mice per group. (**H**) Schematic diagrams showing that the indicated adeno-associated viruses (AAVs) were injected into mouse PVN. (**I**) Adult male *Oxt^Cre* mice were injected with AAV-hSyn-GFP (control) or AAV-hSyn-DIO-hM3Dq-mCherry (hM3Dq) viruses into the PVN, and were then administered with AOM and DSS. The mice were i.p. injected with clozapine-N-oxide (CNO) every other day for 3 weeks (see also *Figure 1—figure supplement 2D*). Two hours after the final dose of CNO, mice were perfused with 4 % paraformaldehyde (PFA). For control, we carried out double immunofluorescence staining for c-Fos (gray) and Oxt (red). For hM3Dq, immunostaining for c-Fos (green) was performed, and Oxt neurons were identified using hM3Dq-mCherry (red). DAPI staining is in blue. Scale bars, 50 μm. (**J**) The percentage of Oxt^PVN neurons expressing c-Fos. n = 4 mice per group. (**K and L**) Male *Oxt^Cre* mice (2 months of age) were injected with the indicated AAV into PVN, and were then treated with AOM and DSS. Subsequently, mice were i.p. administered with CNO every other day for 3 weeks. The animals were then sacrificed and tumor number (**K**) as well as diameter (**L**) were assessed. n = 6 mice per group. (**M**) The density of PCNA-positive cells in tumor tissues. n = 4 (control) or 3 (hM3Dq) mice. (**N**) The density of TUNEL-positive cells in tumor tissues. n = 3 mice per group. Data are shown as means ± SEM. *p < 0.05, **p < 0.01, ***p < 0.001, two-tailed Student's t-test (**C–G, J–N**).

The online version of this article includes the following source data and figure supplement(s) for figure 1:

**Source data 1.** Source data for Figure 1, panels C-G and J-N.

**Figure supplement 1.** Depletion of oxytocin (Oxt) neurons increases anxiety level and promotes colitis-associated cancer (CAC) development in mice.

**Figure supplement 2.** Excitation of Oxt^PVN neurons inhibits colitis-associated cancer (CAC) progression.

**Figure supplement 3.** Density of immune cells in tumor tissues.

*1A*). In the elevated plus maze test, lesion of Oxt neurons decreased the time spent in the open arms (*Figure 1—figure supplement 1B*). Moreover, in the light/dark box test, depletion of Oxt neurons significantly shortened the time spent in the light box (*Figure 1—figure supplement 1C*). Thus, lesion of Oxt neurons elevates anxiety level in mice.

Next, we administered azoxymethane (AOM) and dextran sodium sulfate (DSS) into the adult male *Oxt^Cre* and *Oxt^Cre;DTA* mice to induce colitis-associated cancer (CAC) in the colon and rectum (*Figure 1—figure supplement 1D*). Depletion of Oxt neurons did not significantly impact the body weight or food intake in mice fed a normal chow diet (*Figure 1—figure supplement 1E,F*). After the treatment, colorectal tissues and plasma samples were collected. Indeed, the plasma Oxt levels in *Oxt^Cre;DTA* mice were barely detectable (*Figure 1—figure supplement 1G*), suggesting the disruption of Oxt-producing neurons. Notably, the number and diameter of CAC were both increased in the *Oxt^Cre;DTA* mice (*Figure 1D and E*; *Figure 1—figure supplement 1H*), while colorectal length was not significantly affected (*Figure 1—figure supplement 1I*). Depletion of Oxt neurons promoted cell proliferation in the CAC, as demonstrated by the increased number of cells positive for proliferating cell nuclear antigen (PCNA), a marker for proliferating cell (*Figure 1F*; *Figure 1—figure supplement 1J*). Moreover, lesion of Oxt neurons inhibited cell apoptosis in the tumors as revealed by the reduced number of cells positive for terminal deoxynucleotidyl transferase dUTP nick end labeling (TUNEL) (*Figure 1G*; *Figure 1—figure supplement 1K*). Together, these data indicate that depletion of Oxt neurons promotes CAC development in mice.

Given that depletion of Oxt neurons elevated anxiety level in mice, and that the dysregulation of hypothalamic-pituitary-adrenal (HPA) axis can elicit stress, next, we assessed the circulating adrenocorticotropin (ACTH) and corticosterone levels in *Oxt^Cre* and *Oxt^Cre;DTA* mice with AOM/DSS-induced CAC. Plasma ACTH and corticosterone levels were evidently increased in the *Oxt^Cre;DTA* mice comparing with the *Oxt^Cre* mice (*Figure 1—figure supplement 1L,M*). Thus, the dysregulation of the HPA axis may contribute to the CAC development in the *Oxt^Cre;DTA* mice.

## Chemogenetic activation of Oxt^PVN neurons suppresses CAC progression

Next, we asked whether stimulation of Oxt neurons in the PVN (Oxt^PVN) inhibits CAC progression. To do so, we employed the designer receptor exclusively activated by designer drug (DREADD) (*Roth, 2016*; *Smith et al., 2016*) approach to manipulate these neurons. Specifically, *Oxt^Cre* mice were bilaterally injected with adeno-associated virus (AAV) carrying GFP (AAV-hSyn-GFP), or

Cre-dependent hM3Dq-mCherry into the PVN (*Figure 1H*). To validate the DREADD system, CAC was induced in virus-injected mice. These animals were then intraperitoneally (i.p.) administered with a synthetic ligand, clozapine-N-oxide (CNO) every other day for 3 weeks. Two hours after the final dose of CNO, the mice were perfused with 4 % paraformaldehyde (PFA), and then brain tissues were harvested. Immunofluorescent staining showed that treatment with CNO elicited a robust c-Fos expression in the Oxt^PVN neurons of hM3Dq AAV-injected mice compared with the controls (*Figure 1I and J*), suggesting the activation of these neurons. Mirrored with the results of Oxt neuron depletion, activation of Oxt^PVN neurons significantly relieved anxiety-like behavior in mice (*Figure 1—figure supplement 2A-C*). Thereafter, control and hM3Dq-mCherry AAVs were injected into the PVN of *Oxt^Cre* mice. CAC was induced in these mice using AOM and DSS, and then CNO was i.p. administered every other day for 3 weeks (*Figure 1—figure supplement 2D*). After the treatment, plasma Oxt level was elevated, whereas body weight and food intake had not been significantly affected in hM3Dq AAV-infected mice (*Figure 1—figure supplement 2E,F*). Notably, the elevation of plasma Oxt level following chemogenetic excitation of Oxt neurons has been observed previously (*Grund et al., 2019*). Both tumor number and tumor diameter were reduced in mice whose Oxt^PVN neurons had been excited (*Figure 1K and L*; *Figure 1—figure supplement 2G*), whereas colorectal length was not impacted (*Figure 1—figure supplement 2H*). In agreement with the reduction in tumor size, the number of proliferating cells, revealed by the immunostaining for PCNA, was significantly decreased in hM3Dq AAV-injected mice compared with the controls (*Figure 1M*; *Figure 1—figure supplement 2I*). Besides, the TUNEL assay showed that the number of apoptotic cells was evidently increased (*Figure 1N*; *Figure 1—figure supplement 2J*). Thus, activation of Oxt^PVN neurons inhibits CAC progression by suppressing cell proliferation and promoting cell apoptosis.

Our assays indicated that plasma ACTH and corticosterone levels were markedly decreased in the hM3Dq AAV-injected mice (*Figure 1—figure supplement 2K,L*), implying that the reduced activity of HPA axis may contribute to the tumor suppression effect of Oxt^PVN neuron activation.

The activation of the anti-tumor immunity is crucial for cancer treatment, hence, we asked whether any of the immune cells contributes to the anti-tumor effect of Oxt^PVN neuron activation. To address this question, we assessed these cells in the tumor tissues. Indeed, the number of CD8^+ T cells was markedly increased in hM3Dq AAV-injected mice compared with controls (*Figure 1—figure supplement 3A,F*), and there was no significant change in CD4^+ T cells, B cells, NK cells, or macrophages (*Figure 1—figure supplement 3B-E, and G-J*). Hence, activation of Oxt^PVN neurons may enhance the anti-tumor immunity by increasing the number of CD8^+ T cells.

## The anti-tumor effect of Oxt^PVN neuron activation is dependent on its action in the CNS

Oxt neurons regulate peripheral physiology via both the neural and the endocrine pathways (*Zhang et al., 2021*). Next, we asked whether the central action is important for Oxt^PVN neuron activation to suppress CAC progression. To this end, we elected to centrally block Oxt receptor using L-368,899, an Oxt receptor (OTR) antagonist. Specifically, adult male *Oxt^Cre* mice were bilaterally injected with control or hM3Dq AAV into the PVN, and then CAC was induced using AOM and DSS. Subsequently, these mice were i.p. administered with CNO and i.c.v. injected with aCSF (artificial cerebrospinal fluid) or L-368,899 every other day for 3 weeks (*Figure 2—figure supplement 1A*). After the treatment, these mice were perfused with 4 % PFA, and then brain tissues were sectioned. Immunofluorescent staining showed that treatment with CNO elicited a dramatic c-Fos expression in the Oxt^PVN neurons of hM3Dq AAV-injected mice compared with the controls (*Figure 2A and B*), suggesting the excitation of Oxt^PVN neurons.

Treatment with CNO and L-368,899 did not significantly impact the body weight or food intake in mice (*Figure 2—figure supplement 1B,C*). As anticipated, activation of Oxt^PVN neurons inhibited CAC progression in mice (*Figure 2C–E*). Notably, brain treatment with L-368,899 significantly abrogated this effect (*Figure 2C–E*). Colorectal length remained not impacted in the mice administered with CNO and L-368,899 (*Figure 2F*). Moreover, the immunostaining for PCNA revealed that excitation of Oxt^PVN neurons inhibited cell proliferation, however, this effect was markedly attenuated when the mice were administered with L-368,899 (*Figure 2G and H*). Furthermore, the TUNEL assay showed that the effect of activation of Oxt^PVN neurons on cell apoptosis was diminished when the mice

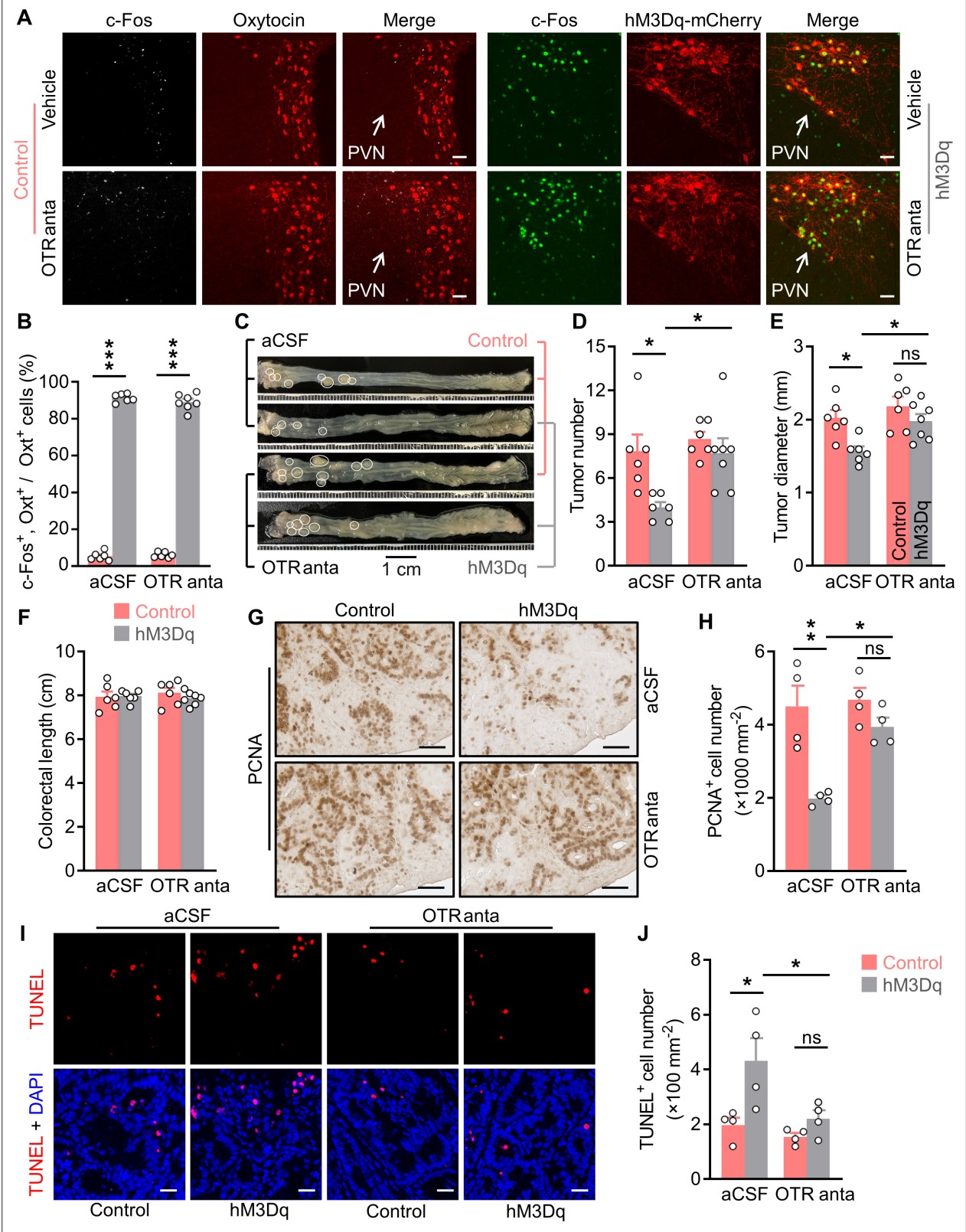

**Figure 2.** Brain oxytocin (Oxt) receptor is crucial for Oxt[PVN] neuron activation to suppress colitis-associated cancer (CAC). (**A**) Adult male *Oxt[Cre]* mice (2 months of age) were injected with AAV-hSyn-GFP (control) or AAV-hSyn-DIO-hM3Dq-mCherry (hM3Dq) viruses into the paraventricular nucleus (PVN), and then colitis-associated cancer (CAC) was induced using azoxymethane (AOM) and dextran sodium sulfate (DSS). Subsequently, these mice were administered with clozapine-N-oxide (CNO) (i.p.), and artificial cerebrospinal fluid (aCSF) or L-368,899 (i.c.v.), an Oxt receptor antagonist (OTR

*Figure 2 continued on next page*

*Figure 2 continued*

anta), every other day for 3 weeks. Mice were then perfused with 4 % paraformaldehyde (PFA). For control, double immunofluorescence staining for c-Fos (gray) and Oxt (red) was performed. For hM3Dq, immunofluorescent staining for c-Fos (green) was performed, and Oxt neurons were identified using hM3Dq-mCherry (red). Cell nuclei were counterstained with DAPI (blue). Scale bars, 50 μm. (**B**) The percentage of Oxt neurons expressing c-Fos in the PVN. n = 7 (hM3Dq, OTR anta) or 6 (all other groups) mice per group. (**C**) The $Oxt^{Cre}$ mice (2 months of age) were injected with indicated adeno-associated viruses (AAVs) into the PVN, and then CAC was induced using AOM and DSS. Subsequently, these mice were administered with CNO (i.p.), as well as aCSF or L-368,899 (i.c.v.), the OTR antagonist (OTR anta), every other day for 3 weeks (see also *Figure 2—figure supplement 1A*). Representative images of colorectal tissue after the treatments are shown. White eclipse outlines the individual tumor. (**D and E**) Tumor number (**D**) and diameter (**E**). ns, not significant. n = 7 (hM3Dq, OTR anta) or 6 (all other groups) mice per group. (**F**) Colorectal length. n = 7 (hM3Dq, OTR anta) or 6 (all other groups) mice per group. (**G and H**) Immunohistochemical staining for proliferating cell nuclear antigen (PCNA) of tumor tissues. Representative images (**G**) and the density of PCNA-positive cells (**H**) are shown. Scale bars, 50 μm. ns, not significant. n = 4 mice per group. (**I and J**) Terminal deoxynucleotidyl transferase dUTP nick end labeling (TUNEL) assay of tumor tissues. Representative images (**I**) and the density of TUNEL-positive cells (**J**) are shown. TUNEL labeling is in red. Cell nuclei were counterstained with DAPI (blue). Scale bars, 20 μm. ns, not significant. n = 4 mice per group. Data are presented as means ± SEM. *p < 0.05, **p < 0.01, ***p < 0.001, one-way ANOVA with Bonferroni's post hoc test.

The online version of this article includes the following source data and figure supplement(s) for figure 2:

**Source data 1.** Source data for Figure 2, panels B, D-F, H and J.

**Figure supplement 1.** Body weight and food intake in mice.

were administered with L-368,899 (*Figure 2I and J*). Collectively, these data suggest that the tumor suppressive effect of $Oxt^{PVN}$ neuron activation is dependent on its action in the CNS.

## $Oxt^{PVN}$ neurons regulate the neuronal activities in the sympathetic CG-SMG

The CNS is known to control peripheral physiology via both the sympathetic nervous system (SNS) and the parasympathetic nervous system (PNS). Besides, the sympathetic celiac-superior mesenteric ganglion (CG-SMG) predominantly innervates colon and rectum. Hence, we examined the effect of $Oxt^{PVN}$ neuron activation on CG-SMG neuronal activity. To do this, adult male $Oxt^{Cre}$ mice were injected with control and hM3Dq AAV into the PVN. After recovery, these mice were i.p. administered with CNO. Two hours later, CG-SMG was dissected and fixed in 4 % PFA. Double immunofluorescence staining for c-Fos and tyrosine hydroxylase (TH), a marker of catecholamine neuron, revealed that the activities of the sympathetic neurons in CG-SMG were significantly inhibited following the activation of $Oxt^{PVN}$ neurons (*Figure 3A and B*).

To confirm this $Oxt^{PVN}$ neuron -> $TH^{CG-SMG}$ neuron pathway, we cut the preganglionic nerve fiber of CG-SMG, and then assessed the neuronal activity in this ganglion using in vivo single-unit recordings. Specifically, adult male $Oxt^{Cre}$ mice were injected with control and hM3Dq AAV into the PVN, and were also implanted with infusion cannula directed to the third ventricle. After recovery, these animals were performed sham operations, or the transection of the preganglionic fiber of CG-SMG (*Figure 3—figure supplement 1A-C*). Subsequently, the 6 min control (1 % DMSO in aCSF) spiking activity was acquired before CNO (1 μg per mouse) application through the pre-implanted cannula. Single-unit spikes from 30 (sham) and 34 (transection) CG-SMG neurons were isolated, and the firing rates were compared before and after CNO infusion (*Figure 3—figure supplement 1D*). Group data showed that i.c.v. administration of CNO significantly reduced the firing frequency of CG-SMG neurons, however, transection of preganglionic fiber significantly abolished this effect (*Figure 3—figure supplement 1E*). Scatterplot of mean firing frequency of individual CG-SMG neuron revealed a mixed modulation following $Oxt^{PVN}$ neurons activation (*Figure 3—figure supplement 1F,G*). The majority of CG-SMG neurons (67%) displayed a decreased firing frequency after CNO infusion. Only a small proportion of neurons (16%) showed an increased firing frequency. The remainder (17%) maintained their activity level after CNO infusion. Yet, after the transection of the preganglionic fiber, the majority of CG-SMG neurons (65%) maintained their activity level after CNO infusion. Hence, following $Oxt^{PVN}$ neuron activation, the signal that leads to the suppression of CG-SMG neurons is transmitted through the preganglionic fiber.

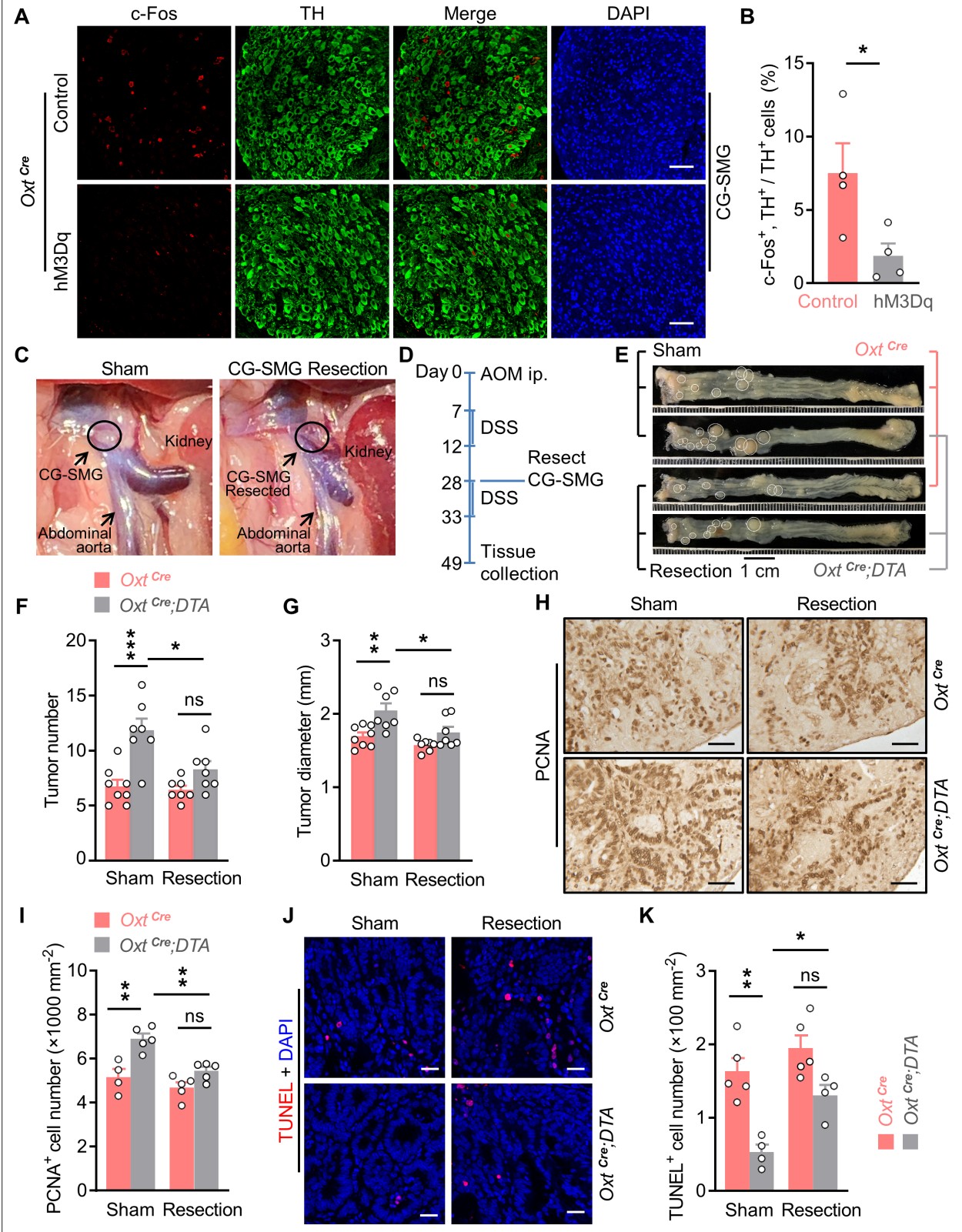

**Figure 3.** Surgical removal of celiac-superior mesenteric ganglion (CG-SMG) attenuates the tumor-promoting effect of oxytocin (Oxt) neuron depletion. (**A**) Adult male *Oxt*$^{Cre}$ mice were injected with control or hM3Dq adeno-associated viruses (AAVs) (hM3Dq) into the paraventricular nucleus (PVN). After surgical recovery, these mice were administered with clozapine-N-oxide (CNO). Two hours later, CG-SMG were dissected and fixed in 4 % paraformaldehyde (PFA). Double immunofluorescence staining for c-Fos (red) and tyrosine hydroxylase (TH, in green) of the CG-SMG was performed.

*Figure 3 continued on next page*

*Figure 3 continued*

Cell nuclei were counterstained with DAPI (blue). Scale bars, 50 μm. (**B**) The percentage of TH-positive cells expressing c-Fos in the CG-SMG. n = 4 mice per group. (**C**) Representative images showing mouse abdominal cavity with (left panel), or without (right panel) CG-SMG (following the resection). (**D**) A schematic diagram of experimental design. The colitis-associated cancer (CAC) was induced in adult *Oxt^Cre* and *Oxt^Cre;DTA* mice using azoxymethane (AOM) and dextran sodium sulfate (DSS). After the first cycle of DSS treatment, sham operation and CG-SMG resection were performed in mice. (**E**) Representative images of colorectal tissue after the treatment. White eclipse was used to outline the individual tumor. (**F and G**) Tumor number (**F**) and diameter (**G**). **ns**, not significant. n = 8 (*Oxt^Cre*, sham) or 7 (all other groups) mice per group. (**H and I**) Immunohistochemical staining for proliferating cell nuclear antigen (PCNA) of tumor tissue. Representative images (**H**) and the density of PCNA-positive cells (**I**) are shown. ns, not significant. Scale bars, 50 μm. n = 4 (*Oxt^Cre*, sham) or 5 (all other groups) mice per group. (**J and K**) Terminal deoxynucleotidyl transferase dUTP nick end labeling (TUNEL) assay of tumor tissue. Representative images (**J**) and the density of TUNEL-positive cells (**K**) are shown. TUNEL labeling is in red. Cell nuclei were counterstained with DAPI (blue). ns, not significant. Scale bars, 20 μm. n = 5 (*Oxt^Cre*) or 4 (*Oxt^Cre;DTA*). Data are presented as means ± SEM. *p < 0.05, **p < 0.01, ***p < 0.001, two-tailed Student's t-test (**B**) or one-way ANOVA with Bonferroni's post hoc test (**F, G, I, K**).

The online version of this article includes the following source data and figure supplement(s) for figure 3:

**Source data 1.** Source data for Figure 3, panels B, F, G, I and K.

**Figure supplement 1.** Transection of the preganglionic fiber of CG-SMG abolishes the inhibitory effect of Oxt neuron activation.

**Figure supplement 2.** Body weight and food intake in mice.

## The CG-SMG is required for lesion of Oxt neurons to promote CAC development

Next, we assessed the Oxt^PVN neuron -> TH^CG-SMG neuron connection using the CAC mouse model. To this end, CAC was induced in the adult *Oxt^Cre* and *Oxt^Cre;DTA* mice using AOM and DSS. After the first cycle of DSS treatment, CG-SMG resection and sham surgeries were performed in mice (*Figure 3C and D*). These manipulations did not significantly impact body weight or food intake in mice (*Figure 3—figure supplement 2A,B*). While depletion of Oxt neurons led to the increasing of CAC number and diameter, CG-SMG resection markedly attenuated these effects (*Figure 3E–G*). We noted that colorectal length was not affected in these mice (*Figure 3—figure supplement 2C*). In agreement with the data of tumor number and size, the effects on cell proliferation and cell apoptosis were both attenuated when CG-SMG were removed from these mice (*Figure 3H–K*). Taken together, the promotion of CAC development owing to Oxt neuron deficiency is mediated by the sympathetic CG-SMG.

## Celastrol enhances Oxt^PVN neuron excitability by increasing their input resistance

Celastrol is a pentacyclic triterpenoid initially extracted from the root of thunder god vine. A recent study showed that treatment with celastrol decreased the body weight in obese mice, but not mice with normal weight (*Ma et al., 2015*). A following study suggested that hypothalamus is critical for celastrol to regulate energy balance (*Liu et al., 2015*). Therefore, we assessed the effect of i.c.v. administered celastrol on hypothalamic neuronal activity. The data showed that the number of c-Fos-positive cells was increased in the PVN, but not other nuclei (*Figure 4—figure supplement 1A,B*), suggesting that brain treatment with celastrol stimulates neurons in the PVN. Oxt neurons in the PVN play a critical role in energy balance control, therefore, we asked whether its activity is modulated by celastrol. To answer this question, we analyzed Oxt neuron excitability after bath application of celastrol via slice electrophysiology. The hypothalamic slices were obtained from *Oxt^Cre;Rosa26-LSL-EYFP* (*Oxt^Cre;EYFP*) mice, in which enhanced yellow fluorescent protein (EYFP) was expressed in Oxt neurons (*Figure 4A*). In response to 500 ms current steps, Oxt neurons fired more action potentials (AP) across increasing current injections in celastrol condition, suggesting an enhanced neuronal excitability (*Figure 4B and C*). We also analyzed the AP waveforms, and found that celastrol increased the size of afterhyperpolarization (*Figure 4D and E*), but did not impact AP threshold, AP amplitude, AP half-width, or AP area (*Figure 4D and F*; *Figure 4—figure supplement 1C-E*). Moreover, celastrol increased input resistance of Oxt neurons, which might increase neuronal excitability (*Figure 4G*). These data implicate that celastrol enhances Oxt neuron firing.

Besides, the above data suggested that celastrol might promote Oxt release from the Oxt^PVN neurons. To address this possibility, we carried out an ex vivo Oxt release assay. The PVN slices were dissected from the male adult C57 BL/6 mice. These tissue slices were balanced in normal Locke's

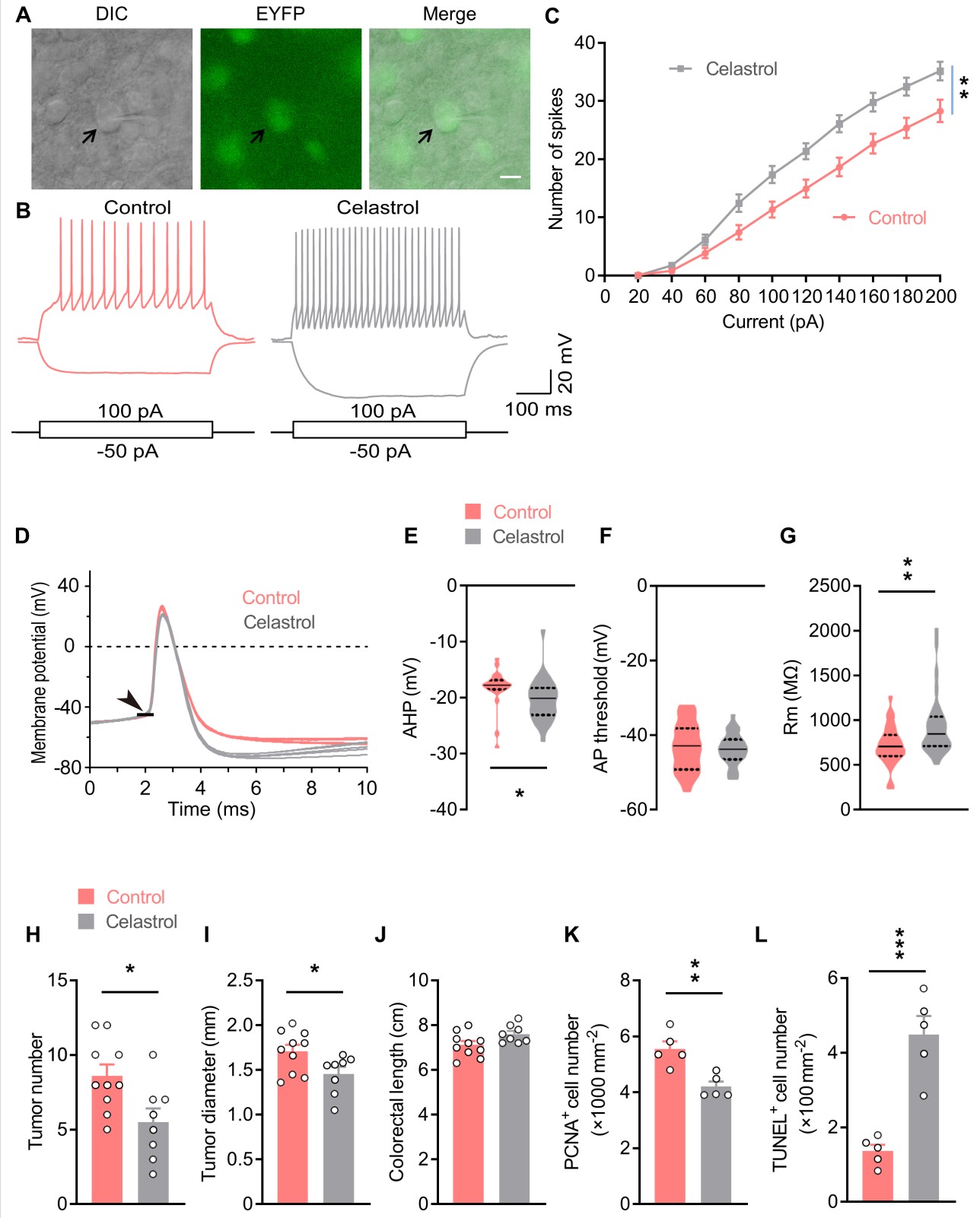

**Figure 4.** Celastrol enhances the excitability of Oxt[PVN] neurons, and its administration in the brain inhibits colitis-associated cancer (CAC) progression. (**A**) Electrophysiology of paraventricular nucleus (PVN) slice of 4 -month-old *Oxt[Cre];EYFP* mice. Left, a differential interference contrast (DIC) image of the recorded neuron (arrow). Middle, expression of enhanced yellow fluorescent protein (EYFP) (green) in the same cell suggests that it is an oxytocin (Oxt) neuron. Right, merged image. Scale bar, 10 μm. (**B**) Voltage response of Oxt neuron in response to 100 and –50 pA current injection in control and

*Figure 4 continued on next page*

*Figure 4 continued*

celastrol (5 µM in artificial cerebrospinal fluid [aCSF]) conditions. (**C**) Bath application of celastrol increased the number of action potentials (AP) fired across increasing current injections. n = 20 cells from five mice (control or celastrol). (**D**) Representative AP traces from control and celastrol conditions. Arrowhead indicates the AP threshold. (**E–G**) The size of afterhyperpolarization (AHP) (**E**), AP threshold, (**F**) and input resistance (Rm) (**G**) in control and celastrol conditions. Solid and dotted lines indicate medians and quartiles, respectively. n = 23 cells (**E, F**) or 27 cells (**G**) from five mice (control) or 28 cells from five mice (celastrol). (**H and I**) The CAC was induced in male C57 BL/6 mice (2 months of age) using azoxymethane (AOM) and dextran sodium sulfate (DSS). These animals were then i.c.v. administered with control versus celastrol every other day for 3 weeks. After the treatment, tumor number (**H**) and diameter (**I**) were determined. n = 10 (control) or 8 (celastrol) mice per group. (**J**) Colorectal length. n = 10 (control) or 8 (celastrol) mice per group. (**K**) The density of proliferating cell nuclear antigen (PCNA)-positive cells in tumor tissue. n = 5 mice per group. (**L**) The density of terminal deoxynucleotidyl transferase dUTP nick end labeling (TUNEL)-positive cells in tumor tissue. n = 5 mice per group. Data are presented as means ± SEM (**C, H–L**). *p < 0.05, **p < 0.01, ***p < 0.001, two-way ANOVA with Bonferroni's post hoc test (**C**), or two-tailed Student's t-test (**E, G, H, I, K, L**).

The online version of this article includes the following source data and figure supplement(s) for figure 4:

**Source data 1.** Source data for Figure 4, panels C and E-L.

**Figure supplement 1.** Celastrol excites neurons in the PVN and promotes Oxt release from PVN.

**Figure supplement 2.** Brain treatment with celastrol suppresses colitis-associated cancer (CAC) progression in mice.

solution, and then in the same solution supplemented with celastrol. The data showed that treatment with celastrol enhanced the rate of Oxt releasing (*Figure 4—figure supplement 1F*). Altogether, these data demonstrate that celastrol could excite Oxt$^{PVN}$ neurons.

## Brain treatment with celastrol suppresses CAC progression in mice

Next, we assessed the effect of brain administered celastrol on CAC progression. To this end, CAC was induced in adult male C57 BL/6 mice using AOM and DSS (*Figure 4—figure supplement 2A*). These mice were then implanted with a guide cannula directed to the third ventricle. After surgical recovery, vehicle and celastrol were administered into the third ventricle via the pre-implanted cannula every other day for 3 weeks (*Figure 4—figure supplement 2A*). Mice receiving celastrol treatment exhibited higher plasma Oxt level than that of the controls (*Figure 4—figure supplement 2B*), suggesting that this chronic treatment stimulated Oxt$^{PVN}$ neurons. Consistent with the previous study (*Liu et al., 2015*), treatment with celastrol did not impact energy balance in CAC mice with normal body weights (*Figure 4—figure supplement 2C,D*). This treatment significantly reduced tumor number and diameter (*Figure 4H,I*; *Figure 4—figure supplement 2E*), while it did not affect colorectal length (*Figure 4J*). Besides, cell proliferation was suppressed, and cell apoptosis was enhanced in the tumor tissue of mice treated with celastrol (*Figure 4K and L*; *Figure 4—figure supplement 2F,G*). Collectively, these data indicate that brain treatment with celastrol suppresses CAC progression in mice.

## Depletion of Oxt neuron abolishes the anti-tumor effect of celastrol

The above data suggested that hypothalamic Oxt neurons are important for celastrol to suppress CAC progression in mice. To address this question, the CAC was induced in the *Oxt$^{Cre}$* and *Oxt$^{Cre}$;DTA* mice (*Figure 5A*). These mice were then i.p. injected with vehicle versus celastrol every other day for 3 weeks (*Figure 5A*). Treatment with celastrol did not significantly impact the body weight or food intake in mice (*Figure 5B and C*). While celastrol inhibited CAC progression in mice, lesion of Oxt neurons could markedly abrogate this effect (*Figure 5D–F*). Lesion of Oxt neuron or celastrol treatment did not have noticeable effect on colorectal length (*Figure 5G*). Notably, the effects of celastrol on cell proliferation and cell apoptosis in CAC were both attenuated in the mice deficient for Oxt neurons (*Figure 5H–K*). Thus, hypothalamic Oxt neurons are required for celastrol to suppress CAC progression.

## Agonism of β2-adrenergic receptor attenuates the anti-tumor effect of Oxt$^{PVN}$ neuron activation

Next, we interrogated whether activation of SNS target, that is, β2 adrenergic receptor (β2AR), would attenuate the anti-tumor effect of Oxt$^{PVN}$ neuron excitation. Our data showed that isoprenaline, an agonist for β2AR, did not affect the activity of CG-SMG neurons (*Figure 6—figure supplement 1A,B*), suggesting that it is proper to use this drug to target CAC cells. Thereafter, adult male *Oxt$^{Cre}$* mice were bilaterally injected with control and hM3Dq AAV into the PVN, and then CAC was induced.

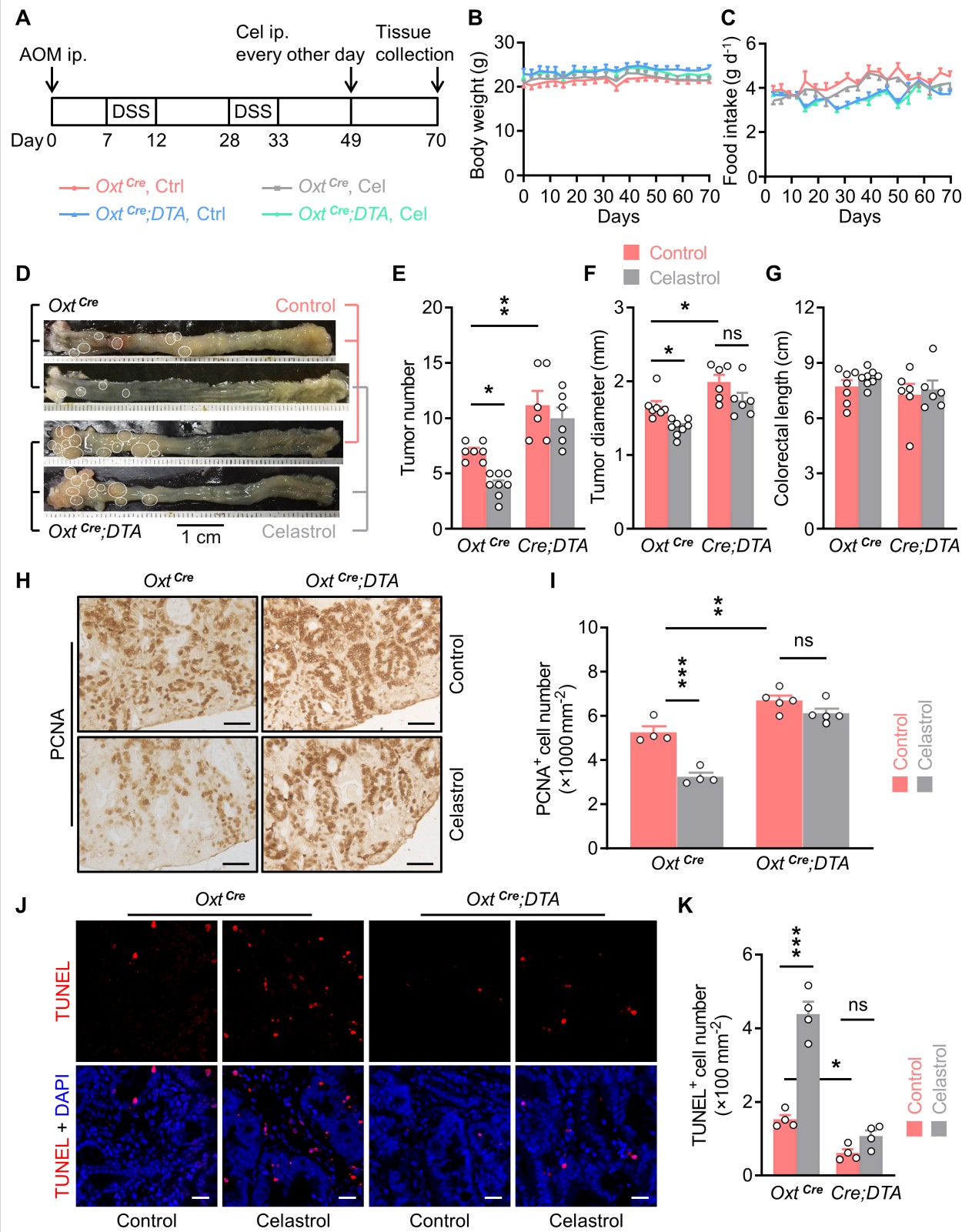

**Figure 5.** Depletion of oxytocin (Oxt) neurons attenuates the anti-tumor effect of celastrol. (**A**) A schematic diagram of experimental design. The colitis-associated cancer (CAC) was induced in the $Oxt^{Cre}$ and $Oxt^{Cre};DTA$ mice (2 months of age), in which control solution and celastrol (Cel) were i.p. administered every other day for 3 weeks. (**B and C**) Body weight (**B**) and food intake (**C**) in mice throughout the experiment. n = 7 ($Oxt^{Cre}$, Ctrl), 8 ($Oxt^{Cre}$, Cel), or 6 ($Oxt^{Cre};DTA$) mice per group. (**D**) Representative images of colorectal tissue after the indicated treatments. White eclipses indicate individual

*Figure 5 continued on next page*

Figure 5 continued

tumor. (**E and F**) Tumor number (**E**) and diameter (**F**). *Cre;DTA*, *Oxt*<sup>Cre</sup>*;DTA*. ns, not significant. n = 7 (*Oxt*<sup>Cre</sup>, Ctrl), 8 (*Oxt*<sup>Cre</sup>, Cel), or 6 (*Oxt*<sup>Cre</sup>*;DTA*) mice per group. (**G**) Colorectal length. n = 7 (*Oxt*<sup>Cre</sup>, Ctrl), 8 (*Oxt*<sup>Cre</sup>, Cel), or 6 (*Oxt*<sup>Cre</sup>*;DTA*) mice per group. (**H and I**) Immunohistochemical staining for proliferating cell nuclear antigen (PCNA) of tumor tissue. Representative images (**H**) and the density of PCNA-positive cells (**I**) are shown. ns, not significant. Scale bars, 50 µm. n = 4 (*Oxt*<sup>Cre</sup>) or 5 (*Oxt*<sup>Cre</sup>*;DTA*) mice per group. (**J and K**) Terminal deoxynucleotidyl transferase dUTP nick end labeling (TUNEL) assay of tumor tissue. Representative images (**J**) and the density of TUNEL-positive cells (**K**) are shown. TUNEL labeling is in red. Cell nuclei were counterstained with DAPI (blue). ns, not significant. Scale bars, 20 µm. n = 4 mice per group. Data are presented as means ± SEM. *p < 0.05, **p < 0.01, ***p < 0.001, one-way ANOVA with Bonferroni's post hoc test (**E, F, I, K**).

The online version of this article includes the following source data for figure 5:

**Source data 1.** Source data for *Figure 5E–G,I,K*.

These mice were i.p. administered with CNO every other day, and were also i.p. injected with saline or isoprenaline on a daily basis. These treatments were continued for 3 weeks (*Figure 6—figure supplement 1C*). Subsequently, these mice were perfused with 4 % PFA, and then brain tissues were sectioned. Immunofluorescent staining showed that treatment with CNO elicited a robust c-Fos expression in the Oxt<sup>PVN</sup> neurons of hM3Dq AAV-injected mice compared with the controls (*Figure 6A and B*), suggesting the activation of these neurons.

Treatment with CNO and/or isoprenaline did not impact the body weight or food intake in control and hM3Dq AAV-injected mice (*Figure 6—figure supplement 1D,E*). Excitation of Oxt<sup>PVN</sup> neurons suppressed CAC progression in mice, however, activation of β2AR with isoprenaline significantly abolished this effect (*Figure 6C–E*). Colorectal length was not significantly impacted in the mice administered with isoprenaline (*Figure 6F*). The histological data revealed that the effects of Oxt<sup>PVN</sup> excitation on cell proliferation and cell apoptosis were dramatically attenuated when isoprenaline was administered (*Figure 6G–J*). Hence, activation of β2AR can significantly abrogate the anti-tumor effect of Oxt<sup>PVN</sup> neuron activation.

## Brain OTR is crucial for centrally administered celastrol to suppress CG-SMG neuronal activity

Our data indicated that Oxt neurons are important for celastrol to restrict CAC development in mice (*Figure 5*). Next, we asked whether i.c.v. administered celastrol could similarly regulate CG-SMG neuronal activity. To address this question, adult male C57 BL/6 mice were implanted with a guide cannula, and were then allowed to recover from surgeries. Subsequently, the preganglionic fiber of CG-SMG was transected, or left intact (sham). These mice were i.c.v. administered with vehicle versus celastrol. Two hours later, CG-SMG was dissected and fixed in 4 % PFA. Double immunofluorescence staining for c-Fos and TH revealed that administration of celastrol suppressed the activity of sympathetic neurons in the CG-SMG. Notably, this effect was markedly diminished when the preganglionic nerve fiber of CG-SMG was transected (*Figure 7—figure supplement 1A-C*).

Thereafter, we asked whether brain OTR is crucial for centrally administered celastrol to suppress the CG-SMG neuronal activity. To this end, adult male C57 BL/6 mice were implanted with a guide cannula directed to the third ventricle. After surgical recovery, these mice were i.c.v. administered with vehicle control or L-368,899, the OTR antagonist, an hour before in vivo single-unit recordings. Subsequently, the 6 min control spiking activity was acquired before celastrol application through the guide cannula (*Figure 7A and B*). Single-unit spikes from 68 CG-SMG neurons (vehicle) and 44 CG-SMG neurons (OTR antagonist) were isolated, and the firing rates were compared before and after celastrol infusion (*Figure 7C and D*). Group data showed that treatment with celastrol significantly reduced the firing frequency of CG-SMG neurons, however, blockade of OTR abrogated this effect (*Figure 7E*). Scatterplot of mean firing frequency of individual CG-SMG neuron revealed a mixed modulation by celastrol (*Figure 7F*). The majority of CG-SMG neurons (63%) displayed a decreased firing frequency after celastrol infusion. Only a small proportion of neurons (18%) showed an increased firing frequency. The remainder (19%) maintained their activity level during celastrol infusion. However, when L-368,899 was applied, the majority of CG-SMG neurons (57%) maintained their activity level during celastrol infusion (*Figure 7G*), suggesting that blockade of OTR could attenuate the inhibitory effect of celastrol on neuronal firing rate in CG-SMG. Together, these data suggest that brain OTR is crucial for centrally administered celastrol to suppress the neuronal activity in the CG-SMG.

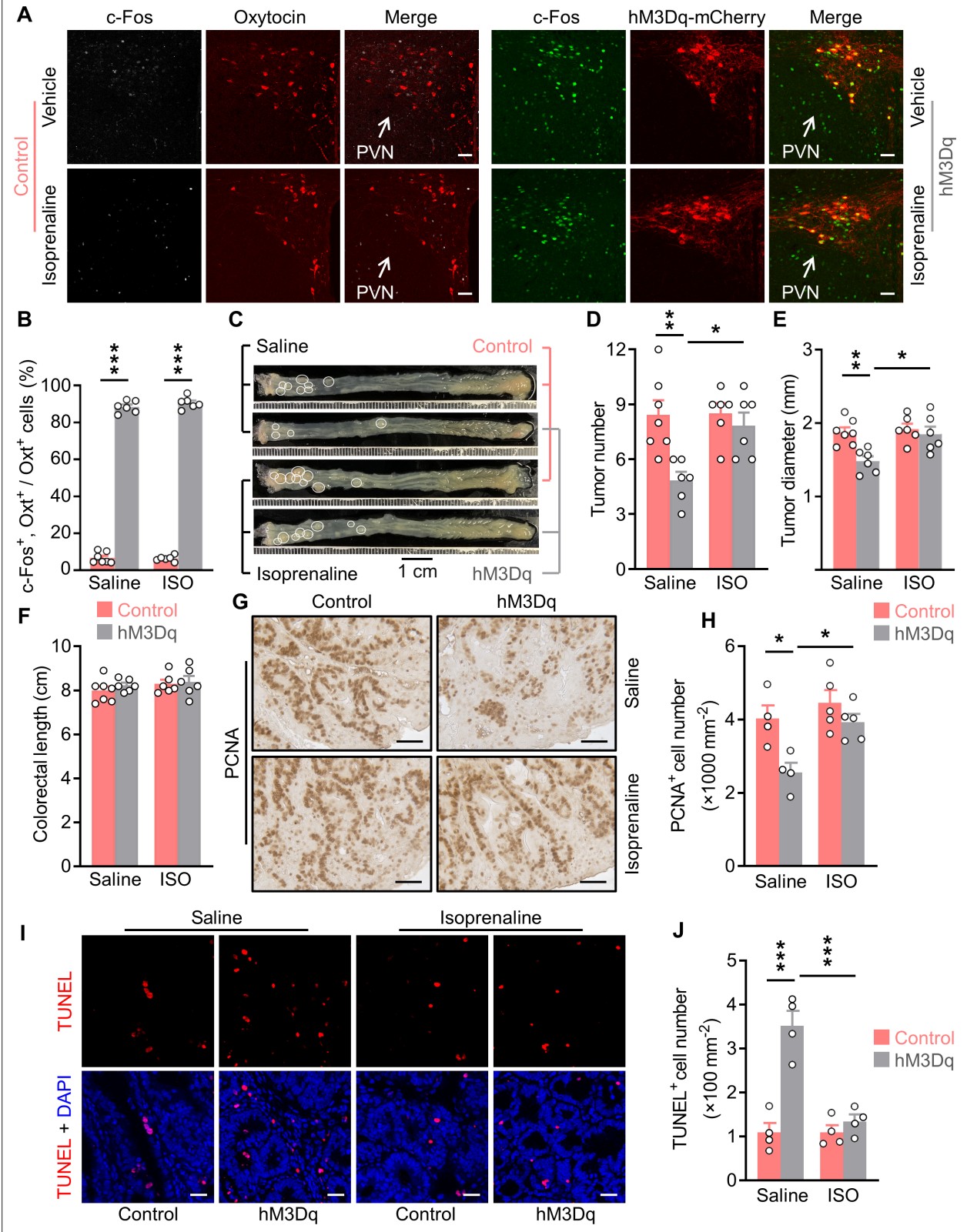

**Figure 6.** Treatment with an agonist for β2 adrenergic receptor attenuates the anti-tumor effect of Oxt$^{PVN}$ neuron activation. (**A**) Control and AAV-hSyn-DIO-hM3Dq-mCherry (hM3Dq) viruses were injected into the paraventricular nucleus (PVN) of male adult Oxt$^{Cre}$ mice. Colitis-associated cancer (CAC) was then induced using azoxymethane (AOM) and dextran sodium sulfate (DSS). These mice were i.p. administered with clozapine-N-oxide (CNO) every other day and i.p. injected with saline or isoprenaline, a β2 adrenergic receptor agonist, on a daily basis. After 3 weeks of treatment, mice were

*Figure 6 continued on next page*

*Figure 6 continued*

perfused with 4 % paraformaldehyde (PFA). For control, double immunofluorescence staining for c-Fos (gray) and oxytocin (Oxt) (red) was performed. For hM3Dq, immunofluorescent staining for c-Fos (green) was performed and Oxt neurons were identified using hM3Dq-mCherry (red). Cell nuclei were counterstained with DAPI (blue). Scale bars, 50 μm. (**B**) The percentage of Oxt neurons expressing c-Fos in the PVN. ISO, isoprenaline. n = 7 (control, saline) or 6 (all other groups) mice per group. (**C**) Adult *Oxt^Cre* mice were injected with adeno-associated viruses (AAVs) into the PVN. CAC was then induced using AOM and DSS. Subsequently, these mice were i.p. administered with CNO every other day and i.p. injected with saline or isoprenaline on a daily basis. These treatments were continued for 3 weeks (see also *Figure 6—figure supplement 1C*). Representative images of colorectal tissue after the treatments are shown. White eclipse outlines individual tumor. (**D and E**) Tumor number (**D**) and diameter (**E**). n = 7 (control, saline) or 6 (all other groups) mice per group. (**F**) Colorectal length. n = 7 (control, saline) or 6 (all other groups) mice per group. (**G and H**) Immunohistochemical staining for proliferating cell nuclear antigen (PCNA) of tumor tissue. Representative images (**G**) and the density of PCNA-positive cells (**H**) are shown. Scale bars, 50 μm. n = 4 (saline) or 5 (ISO) mice per group. (**I and J**) Terminal deoxynucleotidyl transferase dUTP nick end labeling (TUNEL) assay of tumor tissue. Representative images (**I**) and the density of TUNEL-positive cells (**J**) are shown. TUNEL labeling is in red. Cell nuclei were counterstained with DAPI (blue). Scale bars, 20 μm. n = 4 mice per group. Data are presented as means ± SEM. *p < 0.05, **p < 0.01, ***p < 0.001, one-way ANOVA with Bonferroni's post hoc test (**B, D, E, H, J**).

The online version of this article includes the following source data and figure supplement(s) for figure 6:

**Source data 1.** Source data for Figure 6, panels B, D-F, H and J.

**Figure supplement 1.** Body weight and food intake in mice.

## Agonism of β2AR abrogates the tumor suppressive effect of celastrol

Lastly, we interrogated whether the activation of β2AR could attenuate the anti-tumor effect of celastrol. To do so, the AOM/DSS-induced CAC mice were implanted with a guide cannula directed to the third ventricle. After recovery, these animals were i.c.v. administered with vehicle versus celastrol every other day for 3 weeks. Besides, these mice received daily saline or isoprenaline treatment (*Figure 7—figure supplement 2A*). Treatment with celastrol and/or isoprenaline did not impact the body weight or food intake in mice (*Figure 7—figure supplement 2B,C*). As anticipated, brain treatment with celastrol suppressed CAC progression in mice. Yet, treatment with isoprenaline significantly abrogated this effect (*Figure 7H,I*; *Figure 7—figure supplement 2D*). Treatment with celastrol and/or isoprenaline did not impact colorectal length (*Figure 7—figure supplement 2E*). The immunohistochemistry data revealed that treatment with celastrol inhibited cell proliferation, however, this effect was markedly attenuated when the mice were administered with isoprenaline (*Figure 7J*; *Figure 7—figure supplement 2F*). Besides, the TUNEL assay showed that the effect of brain treatment with celastrol on cell apoptosis was diminished when the mice were treated with isoprenaline (*Figure 7K*; *Figure 7—figure supplement 2G*). Overall, these data suggest that activation of β2AR can significantly abolish the anti-tumor effect of centrally administered celastrol.

## Discussion

Negative mood is associated with the occurrences of cancers, however, the underlying mechanisms remain less well understood. In this study, we show that excitation of Oxt^PVN neurons remarkably ameliorated CAC progression in mice, and that this effect was mediated by inhibiting the neuronal activities in the CG-SMG. Also, brain treatment with celastrol suppressed the progression of CAC, and this effect required hypothalamic Oxt neurons. Moreover, we show that β2AR was involved in these processes. Together, our current work demonstrates that modulating hypothalamic Oxt neurons can impact the CAC progression in mice.

Negative moods, such as anxiety, depression, and stress, are implicated in tumor progression. As for CRC, a recent study has revealed a significant association of perceived stress with the incidences of rectal cancer (*Kikuchi et al., 2017*). Perceived stress at work and stressful life events elevated the risk of CRC (*Azizi and Esmaeili, 2015*; *Blanc-Lapierre et al., 2017*). Besides, stress is one of the key contributing factors to the onset and development of spontaneous colitis in humans (*Mitchell and Drossman, 1987*; *Salem and Shubair, 1967*). This association, in particular the one between chronic stress and colitis, was further confirmed in murine models (*Gao et al., 2018*; *Reber et al., 2006*; *Reber et al., 2008*). Moreover, chronic psychosocial stress was shown to result in the deterioration of CAC progression in mice (*Peters et al., 2012*). Hence, these findings suggest that stress is critical for CRC progression. Previous studies showed that Oxt has an anxiolytic effect in both humans (*Heinrichs et al., 2003*) and rodents (*Blume et al., 2008*; *Ring et al., 2006*; *Windle et al., 1997*). Conversely,

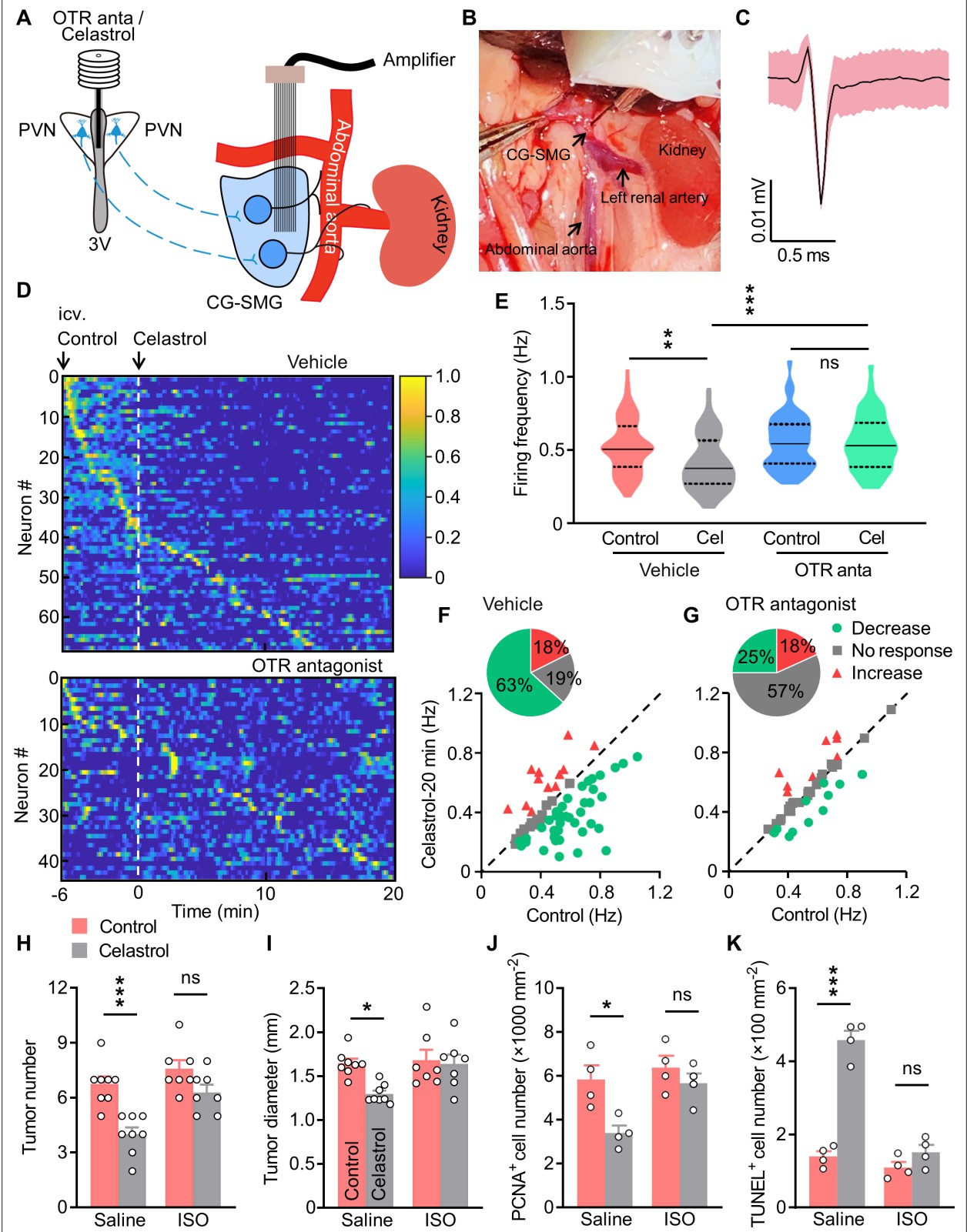

**Figure 7.** Treatment with isoprenaline abolished the anti-tumor effect of celastrol. (**A**) Schematics of in vivo single-unit recordings in celiac-superior mesenteric ganglion (CG-SMG). L-368,899, the Oxt receptor (OTR) antagonist (OTR anta), and celastrol were applied through a guide cannula directed to third ventricle (3 V). (**B**) A CG-SMG image was taken during the operation. (**C**) Example waveform of the single unit detected. (**D**) Normalized firing rate of recorded CG-SMG neurons in response to celastrol infusion in vehicle (top) and OTR antagonist (bottom) groups. Dashed line indicates the

*Figure 7 continued on next page*

**Figure 7 continued**

time point of celastrol delivery. i.c.v., intracerebroventricular injection. n = 68 cells (vehicle) from 7 mice or 44 cells (OTR antagonist) from 6 mice . (**E**) Statistics of average firing frequency of CG-SMG neurons in response to celastrol infusion in vehicle and OTR antagonist groups. Solid and dotted lines indicate medians and quartiles, respectively. n = 68 cells (vehicle) from 7 mice or 44 cells (OTR antagonist) from 6 mice . (**F and G**) Correlation of firing rate before and after celastrol infusion in vehicle (**F**) and OTR antagonist (**G**) groups. Green filled circles represent individual units with significantly lower firing frequency after celastrol infusion. Red triangles represent the units with higher firing rates. Gray squares indicate neurons without significant difference in firing rates. Inset: proportions of CG-SMG neurons with significantly decreased rates, increased rates, or no change in rates after celastrol infusion in vehicle (**F**) and OTR antagonist group (**G**). n = 68 cells (vehicle) from 7 mice or 44 cells (OTR antagonist) from 6 mice . (**H and I**) Colitis-associated cancer (CAC) was induced in male C57 BL/6 mice (2 months of age). These mice were then i.c.v. administered with vehicle (control) or celastrol every other day. In the meantime, the mice were i.p. injected with saline or isoprenaline (ISO) on a daily basis. These treatments were continued for 3 weeks (see also **Figure 7—figure supplement 2A**). Tumor number (**H**) and diameter (**I**) are shown. ns, not significant. n = 8 (saline) or 7 (ISO) mice per group. (**J**) The density of proliferating cell nuclear antigen (PCNA)-positive cells in tumor tissue. ns, not significant. n = 4 mice per group. (**K**) The density of terminal deoxynucleotidyl transferase dUTP nick end labeling (TUNEL)-positive cells in tumor tissue. ns, not significant. n = 4 mice per group. Data are presented as means ± SD (**C**) or means ± SEM (**H–K**). *p < 0.05, **p < 0.01, ***p < 0.001, one-way ANOVA with Bonferroni's post hoc test (**E, H–K**).

The online version of this article includes the following source data and figure supplement(s) for figure 7:

**Source data 1.** Source data for Figure 7, panels E-K.

**Figure supplement 1.** The preganglionic nerve fiber is crucial for brain administered celastrol to suppress neuronal activities in celiac-superior mesenteric ganglion (CG-SMG).

**Figure supplement 2.** Activation of β2 adrenergic receptor abolishes the tumor suppression effect of centrally administered celastrol.

our current and others' previous studies (**Amico et al., 2004**; **Mantella et al., 2003**) demonstrated that disruption of Oxt neuron or *Oxt* gene increased anxiety level in mice. Importantly, we show that depletion of Oxt neuron promoted tumor progression in CAC mice, which agrees with the previous findings showing that increased stress level could promote colorectal tumor progression. Remarkably, our data indicated that chronic excitation of Oxt$^{PVN}$ neurons or treatment with celastrol could significantly inhibit CAC progression in mice. These results are consistent with previous reports displaying that social support reduced the risk of colon cancer (**Ikeda et al., 2013**; **Kinney et al., 2003**). Besides, recent work demonstrated that Oxt has a prosocial role in humans (**Kosfeld et al., 2005**) and rodents (**Lukas et al., 2011**; **Teng et al., 2013**). Altogether, these findings suggest that the anxiolytic property of Oxt is critically important in its anti-tumor effect.

Previous studies unveiled a crucial role for nerve fiber in the tumorigenesis of various organs and tissues. For instance, both the densities of SNS and PNS nerve fibers were correlated with the aggressiveness of human prostate cancer (**Magnon et al., 2013**). Intriguingly, blockade of SNS activity suppressed the development of prostate cancer, whereas blockade of PNS activity inhibited the invasion and metastasis of prostate cancer in mice (**Magnon et al., 2013**). A further study indicated that norepinephrine released from SNS nerves drove angiogenesis in prostate cancer (**Zahalka et al., 2017**). Besides, a recent study showed that vagal innervation contributed to the development of stomach cancer via muscarinic acetylcholine $M_3$ receptor (**Zhao et al., 2014**). Infiltration of nerve fibers was associated with the aggressiveness of breast cancer (**Pundavela et al., 2015**). The sensory neurons were able to facilitate the initiation and progression of pancreatic ductal adenocarcinoma in mice (**Saloman et al., 2016**). Together, these findings underscore an important role for nerve fiber of the autonomous nervous system in the initiation, invasion, or metastasis of cancers in peripheral organs, and hence the term 'cancer neuroscience' was coined (**Demir et al., 2020**; **Monje et al., 2020**). However, whether the CNS is similarly important remains largely unknown. In this work, we show that stimulation of Oxt$^{PVN}$ neurons could suppress CAC progression in mice. Thus, in concert with other evidence (**Cao et al., 2010**; **Liu et al., 2014**), our current study implicates a critical role for the CNS, in particular the hypothalamus, in peripheral tumor development.

In summary, our current study indicates that chemogenetic stimulation of Oxt$^{PVN}$ neurons or brain treatment with celastrol can suppress CAC progression in mice. The anti-tumor effect of celastrol requires hypothalamic Oxt neurons. Overall, these results suggest that modulating Oxt neuronal activity might be a relevant strategy for the treatment of CRC.

# Materials and methods

**Key resources table**

| Reagent type (species) or resource | Designation | Source or reference | Identifiers | Additional information |
|---|---|---|---|---|
| Genetic reagent (*Mus. musculus*) | *Oxt$^{Cre}$* | Jackson Laboratory | 024234 | PMID:23028821 |
| Genetic reagent (*Mus. musculus*) | *Rosa26DTA176* | PMID:16407399 | | |
| Genetic reagent (*Mus. musculus*) | *Rosa26-LSL-EYFP* | PMID:11299042 | | |
| Strain, strain background (*AAV*) | AAV-hSyn-GFP | Obio Technology | AOV062 | |
| Strain, strain background (*AAV*) | pAAV-hSyn-DIO-hM3Dq-mCherry | Obio Technology | HYMBH2482 | |
| Antibody | (Rabbit polyclonal) anti-c-Fos | Santa Cruz Biotechnology | Cat# sc-7202; RRID:AB_2106765 | IF, (1:150) |
| Antibody | (Goat polyclonal) anti-c-Fos | Santa Cruz Biotechnology | Cat# sc-52-G; RRID:AB_2629503 | IF, (1:25) |
| Antibody | (Mouse monoclonal) anti-TH | Santa Cruz Biotechnology | Cat# sc-25269; RRID:AB_628422 | IF, (1:200) |
| Antibody | (Mouse monoclonal) anti-CD4 | Santa Cruz Biotechnology | Cat# sc-19641; RRID:AB_10554681 | IHC, (1:50) |
| Antibody | (Mouse monoclonal) anti-CD11b | Santa Cruz Biotechnology | Cat# sc-53086; RRID:AB_628894 | IHC, (1:100) |
| Antibody | (Rabbit polyclonal) anti-c-Fos | Abcam | Cat# ab190289; RRID:AB_2737414 | IF, (1:2000) |
| Antibody | (Rabbit polyclonal) anti-Oxt | Immunostar | Cat# 20068; RRID:AB_572258 | IF, (1:400) |
| Antibody | (Mouse monoclonal) anti-PCNA | Boster Biological | Cat# BM0104 | IHC, (1:200) |
| Antibody | (Rabbit polyclonal) anti-CD8α | Bioss | Cat# bs-0648R; RRID:AB_10857537 | IHC, (1:250) |
| Antibody | (Rat monoclonal) anti-B220 | BD Biosciences | Cat# 553087; RRID:AB_394617 | IHC, (1:300) |
| Antibody | (Mouse monoclonal) anti-NK1.1 | BD Biosciences | Cat# 550627; RRID:AB_398463 | IHC, (1:400) |
| Commercial assay or kit | Oxytocin EIA kit | Enzo Life Sciences | Cat# ADI-900–153 A; RRID:AB_2815012 | |
| Commercial assay or kit | Corticosterone ELISA kit | Enzo Life Sciences | Cat# ADI-900–097; RRID:AB_2307314 | |
| Commercial assay or kit | ACTH ELISA kit | Aviva Systems Biology | Cat# OKEH00628 | |
| Commercial assay or kit | In Situ Cell Death Detection Kit, TMR red | Sigma-Aldrich | Cat# 12156792910 | |
| Commercial assay or kit | SABC-POD kit | Boster Biological | Cat# SA1021 | |
| Chemical compound, drug | Azoxymethane | Sigma-Aldrich | Cat# A5486 | |
| Chemical compound, drug | Avertin | Sigma-Aldrich | Cat# T48402 | |
| Chemical compound, drug | Isoprenaline | Sigma-Aldrich | Cat# I5627 | |
| Chemical compound, drug | Proteinase K | Sigma-Aldrich | Cat# 3115879001 | |

| Reagent type (species) or resource | Designation | Source or reference | Identifiers | Additional information |
|---|---|---|---|---|
| Chemical compound, drug | Dextran sulfate sodium | TdB Labs | Cat# 9011-18-1 | |
| Chemical compound, drug | CNO | MedChemExpress | Cat# HY-17366 | |
| Chemical compound, drug | Celastrol | Mengry Bio-Technology | Cat# MR80328 | |
| Chemical compound, drug | L-368,899 | Santa Cruz Biotechnology | Cat# sc-204037 | |
| Softwares, algorithm | Pclamp 10 acquisition | Molecular Devices | | |
| Softwares, algorithm | OmniPlex neural recording data acquisition system | Plexon | | |
| Softwares, algorithm | Offline Sorter V4.0 | Plexon | | |
| Softwares, algorithm | Neuroexplorer V5.0 | Plexon | | |
| Softwares, algorithm | Matlab R2019b | MathWorks | | |
| Softwares, algorithm | Photoshop | Adobe | | |
| Softwares, algorithm | Prism 8 | GraphPad Software | RRID:SCR_002798 | |
| Softwares, algorithm | ImageEP software | PMID:19229173 | | |
| Softwares, algorithm | ImageLD software | PMID:18704188 | | |
| Softwares, algorithm | ImageOF software | https://cbsn.neuroinf.jp/modules/xoonips/detail.php?id=ImageOF | | |

## Mice

The $Oxt^{Cre}$ (*Wu et al., 2012*) mouse line was purchased from the Jackson Laboratory (Bar Harbor, ME). $Rosa26^{DTA176}$ (*Wu et al., 2006*) and *Rosa26-LSL-EYFP* (*Srinivas et al., 2001*) mice have been described previously. We generated the $Oxt^{Cre}$;$Rosa26^{DTA176}$ mice by crossing the $Oxt^{Cre}$ with the $Rosa26^{DTA176}$ mice, and the $Oxt^{Cre}$;*Rosa26-LSL-EYFP* ($Oxt^{Cre}$;*EYFP*) mice by crossing the $Oxt^{Cre}$ with the *Rosa26-LSL-EYFP* mice. C57 BL/6 mice were purchased from the Vital River Laboratory Animal Technology (Beijing, China). Rodent chow diet was purchased from HFK Bioscience (Beijing, China). All mice were housed in a 12-hr light/12-hr dark cycle in a temperature-controlled room (22–24°C).

## Antibodies and chemicals

Rabbit and goat anti-c-Fos, mouse anti-TH, anti-CD4, and anti-CD11b antibodies were purchased from Santa Cruz Biotechnology (Santa Cruz, CA). Rabbit anti-c-Fos antibody was purchased from Abcam (Cambridge, UK). Rabbit anti-Oxt antibody was obtained from Immunostar (Hudson, WI). Mouse anti-PCNA antibody was purchased from Boster Biological (Wuhan, China). Rabbit anti-CD8α antibody was purchased from Bioss (Woburn, MA). Rat anti-B220 and mouse anti-NK1.1 antibodies were obtained from BD Biosciences (San Diego, CA). Alexa Fluor (AF) 488 goat anti-rabbit, AF 555 donkey anti-rabbit, AF 633 donkey anti-goat, and AF 488 donkey anti-mouse secondary antibodies were purchased from Thermo Fisher (Waltham, MA).

Azoxymethane, isoprenaline, and Avertin were purchased from Sigma-Aldrich (St Louis, MO). Dextran sulfate sodium was obtained from TdB Labs (Uppsala, Sweden). CNO was purchased from MedChemExpress (Monmouth Junction, NJ). Celastrol was obtained from Mengry Bio-Technology (Shanghai, China). L-368,899 was purchased from Santa Cruz Biotechnology.

## AOM/DSS-induced CAC mouse model

Male mice were i.p. injected with the azoxymethane (12.5 mg kg$^{-1}$). A week later, mice were administrated with two cycles of 5-day oral exposure to DSS (2.5 % in drinking water) and then 16-day normal drinking water (*Neufert et al., 2007*).

## Stereotaxic surgery

Third ventricle cannulation: The procedures have been described before (*Wu et al., 2017*; *Zhang et al., 2008*). Briefly, mice were anesthetized with Avertin (300 mg kg$^{-1}$) and were then placed on an ultra-precise stereotaxic instrument (David Kopf, Tujunga, CA). Next, a guide cannula (RWD Life Science, Shenzhen, China) was placed directed to third ventricle (coordinates: A/P –2.0 mm posterior to bregma, D/V –5.0 mm). Mice were allowed to fully recover from surgeries.

For AAV injection, mice were anesthetized and placed on the stereotaxic instrument. With the help of a guide cannula, viral solution was injected bilaterally into the PVN (coordinates: A/P, –0.85 mm posterior to bregma, M/L, ± 0.2 mm, D/V, –4.8 mm).

## Chemogenetics

AAVs carrying GFP (AAV-hSyn-GFP) or Cre-dependent hM3Dq-mCherry (AAV-hSyn-DIO-hM3Dq-mCherry) were purchased from Obio Technology (Shanghai, China). Adult male *Oxt^Cre^* mice were bilaterally injected with AAVs into the PVN, and were then allowed to recover from surgeries. After the induction of CAC, mice were i.p. administered with CNO (3 mg kg$^{-1}$, every other day for 3 weeks) to activate the hM3Dq-expressing Oxt neurons .

## Treatments

Treatment with CNO and L-368,899: The control and hM3Dq AAVs were injected into the PVN of adult *Oxt^Cre^* mice. CAC was induced using AOM and DSS. These mice were i.p. injected with CNO and i.c.v. administered with vehicle or L-368,899 (2 µg per mouse) every other day for 3 weeks. Body weight and food intake in mice were recorded throughout the experiment.

Celastrol: Adult male C57 BL/6 mice bearing AOM and DSS-induced CAC were implanted with a guide cannula directed to the third ventricle, and were then allowed to recover from surgeries. aCSF and celastrol (0.5 µg per mouse) was i.c.v. administered every other day for 3 weeks. In a separate experiment, adult male and female *Oxt^Cre^* and *Oxt^Cre^;DTA* mice were administered with AOM and DSS to induce CAC, and were then i.p. injected with vehicle (1 % DMSO in saline) or celastrol (1 mg kg$^{-1}$) every other day for 3 weeks. Body weight and food intake were regularly assessed throughout the experiment.

Treatment with CNO and isoprenaline: The control and hM3Dq AAVs were injected into the PVN of male *Oxt^Cre^* mice, in which CAC was then induced. These mice were i.p. administered with CNO (3 mg kg$^{-1}$) every other day for 3 weeks. During this period, saline and isoprenaline (10 mg kg$^{-1}$) were i.p. administered on a daily basis. Body weight and food intake in mice were assessed.

Treatment with celastrol and isoprenaline: Adult male C57 BL/6 mice bearing CAC were i.c.v. administered with vehicle or celastrol (0.5 µg per mouse) every other day for 3 weeks. In the meanwhile, these mice were i.p. injected with saline or isoprenaline (10 mg kg$^{-1}$) on a daily basis. Body weight and food intake in mice were measured.

## Removal of CG-SMG, and the transection of its preganglionic nerve fiber

Mice were anesthetized using Avertin, and then the abdomen was cut open. Abdominal viscera were gently pulled out and held in warm sterile saline-soaked gauze. The intersection of the descending aorta and the left renal artery was identified, where the superior mesenteric artery was located. The CG-SMG is wrapped around the superior mesenteric artery and associated lymphatic vessels. Fine forceps and microdissection scissor were used to remove CG-SMG or transect its preganglionic nerve fiber.

## Slice electrophysiology

The *Oxt^Cre^;EYFP* mice (4 months of age) were euthanized with an overdose of sodium pentobarbital (40 mg kg$^{-1}$, i.p.). Coronal PVN slices (300 µm in thickness) were cut in a solution containing (in mM): 228 sucrose, 26 NaHCO$_3$, 11 glucose, 2.5 KCl, 1 NaH$_2$PO$_4$, 7 MgSO$_4$, and 0.5 CaCl$_2$, and recovered in aCSF containing (in mM): 119 NaCl, 26 NaHCO$_3$, 11 glucose, 2.5 KCl, 1 NaH$_2$PO$_4$, 1.3 MgSO$_4$, and 2.5 CaCl$_2$. Recordings were performed in a submerged-style chamber mounted under an infrared-differential interference contrast microscope (BX-51 WI, Olympus, Tokyo, Japan). Slices were constantly perfused with heated aCSF (35°C) and bubbled continuously with 95 % O$_2$ and 5 % CO$_2$. Oxt neurons were

identified by EYFP epifluorescence. Whole-cell recordings were achieved using a Multiclamp 700B amplifier (Molecular Devices, San Jose, CA). Signals were filtered at 10 kHz, and then sampled by Digidata 1550B4 (Molecular Devices) at 20 kHz using Clampex 10 acquisition software. The pipette resistance was about 4–6 MΩ with an internal solution containing (in mM): 135 K-gluconate, 8 KCl, 10 HEPES, 0.25 EGTA, 2 MgATP, 0.3 $Na_3GTP$, 0.1 spermine, 7 phospho-creatine (pH 7.25–7.3; osmolarity 294–298). For celastrol condition, celastrol (5 µM) was added to the incubation chamber 20 min prior to recording and was added in bath aCSF throughout recording. Liquid junction potential (16 mV) has been corrected in the text and figures.

## In vivo single-unit recordings

Male mice (8 weeks of age) were implanted with a guide cannula directed to the third ventricle. Two weeks later, in vivo single-unit recordings were performed and analyzed as described previously (*Tseng et al., 2011*). The guide tubes housed 16-channel electrodes using 25.4 µm formvar-insulated nichrome wire (761500, A-M System, Sequim, WA). The final impedance of the electrodes was 700–800 kΩ. On the recording day, the CG-SMG located at the intersection of the descending aorta and left renal artery was identified, and the 16-channel electrodes were manually placed into CG-SMG. A sterile cotton swab was dipped in saline solution, and was then placed by the CG-SMG to maintain tissue humidity during recording. Spiking activities were digitized at 40 kHz, bandpass-filtered from 250 to 8000 Hz, and stored on a PC for further offline analysis.

For administration of celastrol and L-368,899, the C57 BL/6 mice were implanted with an infusion cannula directed to third ventricle and were then singly housed to allow recovery from surgeries. On the recording day, aCSF and L-368,899 were applied through the pre-implanted cannula 1 hr before recordings. The 6 min control (5 % DMSO in aCSF) spiking activity was acquired before celastrol (0.5 µg per mouse) application through the infusion cannula.

In the CG-SMG preganglionic nerve fiber transection experiment, adult $Oxt^{Cre}$ mice were injected with control or hM3Dq AAV into the PVN. These mice were also implanted with an infusion cannula directed to third ventricle. After recovery, the preganglionic nerve fiber of CG-SMG was transected before recording. In the control group, sham operations were carried out before recording. Subsequently, the 6 min control (1 % DMSO in aCSF) spiking activity was acquired before CNO (1 µg per mouse) application through the infusion cannula.

## In vivo single-unit recordings data analysis

The single-unit spike sorting was performed with Offline Sorter V4.0 (Plexon, Dallas, TX). Spikes were detected when a minimum waveform reached an amplitude threshold of –4.50 standard deviation greater than the noise amplitude. Principal component analysis and automatic scan were employed to separate single-unit waveforms into individual clusters. Manual checking was then performed to ensure that the cluster boundaries were clearly separated. All isolated single units exhibited recognizable refractory periods (>1 ms) in the inter-spike interval histograms. Only well-isolated units (L ratio <0.2, isolation distance >15) were included in the data analysis.

The response of single unit was analyzed with Neuroexplorer V5.0 (Plexon). Well-separated units were used to analyze the responses before (baseline) and after celastrol or CNO infusion. Firing rates of neurons during baseline, 10 and 20 min after celastrol or CNO infusion were compared to determine the significance of difference in firing rates (paired Student's t-test, 95 % confidence interval). For heatmap analysis, z-score of each bin (10 s) was calculated by the following equation: $z = (x-\mu)/\sigma$, in which x is the raw firing rate, µ is the mean firing rate during the baseline period, and σ is the corresponding standard deviation. Further normalization was utilized for better presentation. All of the single-unit z-scores were plotted using Matlab R2019b (Natick, MA).

## Immunofluorescence

The detailed procedures have been described previously (*Shen et al., 2020*). Mice were anesthetized using Avertin, and were then transcardially perfused with 4 % PFA. Mouse brains were removed, post-fixed in 4 % PFA, and infiltrated with 20–30% sucrose solutions. Brain tissues were sectioned using a cryostat. Tissue sections were washed with phosphate buffered saline (PBS), blocked with 5 % serum/0.3 % Triton X-100/PBS for 30 min, incubated with primary antibodies at 4 °C overnight, and

fluorophore-conjugated secondary antibodies at room temperature for 1 hr. Cell nuclei were counter-stained with DAPI.

Immunofluorescence staining of CG-SMG: Mice were euthanized, and then the CG-SMG were dissected, fixed in 4 % PFA for 10 min. The tissues were infiltrated with 75–100% ethanol, and were then embedded in paraffin and sectioned (thickness: 3 μm). The tissue sections were deparaffinized and rehydrated using graded ethanol. Antigen retrieval was then performed. Tissue sections were washed with 1× PBS, blocked with 5 % serum/0.3 % Triton X-100/PBS for 30 min, incubated with primary antibodies at 4°C overnight, and fluorophore-conjugated secondary antibodies at room temperature for 1 hr. Cell nuclei were counterstained with DAPI. Images were acquired with the LSM 780 confocal microscope (Carl Zeiss, Jena, Germany). Cells were manually counted in one representative image collected for each mouse.

## Immunohistochemistry

Paraffin-embedded tissue sections were deparaffinized, rehydrated, and antigen-recovered. Sections were then blocked with 5 % serum/0.3 % Triton X-100/PBS for 30 min, incubated with primary antibodies at 4°C overnight and followed by a reaction using a SABC-POD kit (Boster Biological). Images were acquired using an IX71 microscope (Olympus). Cells were counted using Photoshop (Adobe, San Jose, CA).

## TUNEL assay

The In Situ Cell Death Detection Kit was purchased from Sigma-Aldrich. Paraffin-embedded tissue sections were deparaffinized and rehydrated. Next, tissue sections were rinsed in distilled water, incubated with proteinase K (18.5 μg ml$^{-1}$ in 10 mM Tris·HCl) at 37°C for 15 min, washed with 1× PBS, and were then incubated with TUNEL reaction mixture in the humidified chamber at 37 °C for 1 hr. Cell nuclei were counterstained with DAPI. Images were acquired with the LSM 780 confocal microscope. TUNEL-positive cells were manually counted using Photoshop.

## Behavioral analyses

Open field test: Adult male *Oxt$^{Cre}$*, *Oxt$^{Cre}$;DTA* mice, and the *Oxt$^{Cre}$* mice injected with control or hM3Dq AAV were placed in an opaque, square open field (40 cm L × 40 cm W × 40 cm H), and were then allowed to freely explore for 5 min and monitored with the ImageOF software (https://cbsn.neuroinf.jp/modules/xoonips/detail.php?id=ImageOF). The open field was divided into a peripheral region and a 13.3 cm × 13.3 cm central region. Time spent in the central versus peripheral region during the test was presented.

Elevated plus maze test: the plus maze had two closed arms (35 cm L × 6 cm W × 22 cm H) and two open arms (35 cm L × 6 cm W). The maze was elevated 74 cm from the floor. Mice were placed on the center section and allowed to explore the maze freely and monitored with ImageEP software (*Komada et al., 2008*). Time spent in the open versus closed arms during the 5 min period was presented.

Light/dark box test: The apparatus was comprised of a dual compartment box (20 cm L × 20 cm W × 40 cm H) with free access between them. The dark box was made of black Plexiglass and the light one was exposed to room light. The exploratory activity was monitored for 5 min using the ImageLD software (*Takao and Miyakawa, 2006*). Time spent in the light versus dark box was presented.

## Oxt release assay

The detailed procedures have been described previously (*Zhang et al., 2011*). In order to determine the effect of celastrol on Oxt release, PVN tissue slices were dissected from the brain of C57 BL/6 mice and were balanced in normal Locke's solution supplied with 95 % O$_2$ and 5 % CO$_2$ at 37 °C. The solution was changed every 5 min for 10 times and the 11th sample was collected to measure the basal Oxt release rate. The slices were then incubated in the same solution containing celastrol (5 μM) for 5 min and this solution was measured to determine the Oxt release rate under celastrol condition. An oxytocin EIA kit (Enzo Life Sciences, Farmingdale, NY) was used to determine the Oxt concentration in the solutions.

## Plasma Oxt, ACTH, and corticosterone assays

The plasma was collected from mice after the completion of the experiments. Plasma Oxt and corticosterone levels were determined using the Oxt EIA kit and a corticosterone ELISA kit (Enzo Life

Sciences), respectively. Plasma ACTH was assessed using an ACTH ELISA kit (Aviva Systems Biology, San Diego, CA).

## Statistical analysis

All data are presented as means ± SEM unless otherwise specified. Sample sizes with sufficient power were determined according to our published studies and relevant literature. Animals were assigned to specific experimental groups without bias. Data were analyzed using Prism 8 (GraphPad Software, San Diego, CA) or Matlab R2019b. Data distribution was assumed to be normal but this was not formally tested. Two-group comparisons were assessed using two-tailed Student's t-test. One-way and two-way analysis of variance (ANOVA) with Bonferroni's post hoc test was used for comparisons of more than two groups. Key experiments were repeated at least twice independently. No data were excluded from the analyses. When necessary, experimental performers were blind to group information before data were obtained. A p-value of less than 0.05 was considered statistically significant.

## Acknowledgements

This work was supported by the National Natural Science Foundation of China (81573146 and 91539125 to GZ, 81972767 to ML, 31871089 to YH, and 31871028 to JM).

## Additional information

### Funding

| Funder | Grant reference number | Author |
| --- | --- | --- |
| National Natural Science Foundation of China | 81573146 | Guo Zhang |
| National Natural Science Foundation of China | 91539125 | Guo Zhang |
| National Natural Science Foundation of China | 81972767 | Mei Liu |
| National Natural Science Foundation of China | 31871089 | Yunyun Han |
| National Natural Science Foundation of China | 31871028 | Man Jiang |

The funders had no role in study design, data collection and interpretation, or the decision to submit the work for publication.

### Author contributions

Susu Pan, Data curation, Formal analysis, Methodology, Visualization, Writing - original draft; Kaili Yin, Shuren Wang, Data curation, Formal analysis, Methodology; Zhiwei Tang, Yirong Wang, Slice electrophysiology and in vivo single-unit recordings; Zhuo Chen, Data curation, Formal analysis; Hongxia Zhu, Formal analysis, Writing – review and editing; Yunyun Han, Formal analysis, Funding acquisition; Mei Liu, Formal analysis, Funding acquisition, Methodology, Writing – review and editing; Man Jiang, Formal analysis, Funding acquisition, Slice electrophysiology and in vivo single-unit recordings, Visualization, Writing – review and editing; Ningzhi Xu, Formal analysis, Resources, Supervision, Writing – review and editing; Guo Zhang, Conceptualization, Funding acquisition, Supervision, Writing – review and editing

### Author ORCIDs

Man Jiang ![ORCID] http://orcid.org/0000-0002-0470-8722
Guo Zhang ![ORCID] http://orcid.org/0000-0002-3880-6996

### Ethics

Animal procedures were approved by the IACUC at Huazhong University of Science and Technology (#2511).

Decision letter and Author response
Decision letter https://doi.org/10.7554/eLife.67535.sa1
Author response https://doi.org/10.7554/eLife.67535.sa2

## Additional files

### Supplementary files
• Transparent reporting form

### Data availability
All data that support the findings of this study are included in this published article and its supplementary files. Source data files have been provided for Figures 1-7.

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
