## [Decision Letter]

**Acceptance summary:**

This manuscript is of broad interest to gastrointestinal physiologists, cancer biologists and neuroscientists, including readers whose interests include the hypothalamic effects of anxiety as well as central effects in cancer. Pan et al., studied the consequences of manipulations of hypothalamic oxytocin (OT) neurons on pharmacologically induced colorectal cancer progression in mice and determined that celastrol, a pentacyclic triterpenoid, which should excite OT neurons, also inhibited colorectal cancer progression, an effect, which was attenuated in OT neuron-depleted mice. The authors used a series of overlapping experimental manipulations (surgical, genetic, chemogenetic, pharmacological and electrophysiological) as a way to identify the role of the sympathetic nervous system in contributing to these effects as well as to dissect circuit and to largely support the key claims of the paper.

**Decision letter after peer review:**

Thank you for resubmitting your work entitled "Stimulation of hypothalamic oxytocin neurons suppresses colorectal cancer progression in mice" for consideration by *eLife*. Your article has been reviewed by 3 peer reviewers, one of whom is a member of our Board of Reviewing Editors and the evaluation has been overseen by Mone Zaidi as the Senior Editor. The following individual involved in the review of your submission has agreed to reveal their identity: Jeff Roizen (Reviewer #3).

The manuscript has been improved but there are some remaining issues that need to be addressed, as outlined below:

Essential revisions:

In order to increase the impact of the paper, and to justify the conclusions drawn, especially to show causalities between OT manipulations and progression in colorectal cancer, the following points should be taken into consideration:

1) The role of the sympathetic nervous system suppression in contributing to these effects is not yet clear. The authors could first identify that sympathetic neurons were impacted using a marker specific to catecholamine neurons (tyrosine hydroxylase, for example). Secondly, the authors could assess the effects in animals that lack SNS innervation to the target tissues in question using either surgical or chemical ablation (6-OHDA) approaches as opposed to a β-2 receptor agonist. Related to this is that the sympathetic nervous system effect was only examined in the context of their novel reagent (rather than, for instance, in the DREADD-dependent model).

2) What is missing is a proposed causal mechanism of the anticancer effect of OT neuron activation. Is it the attenuation of the activity of the HPA axis, as repeatedly shown by OT, is it the reduction in chronic stress levels mediated by OT, are anti-inflammatory effects involved or other effects on the immune system? An experiment blocking OT receptors (centrally or within selected brain regions) or an experiment manipulating corticosterone levels during OT neuronal activation or depletion might be helpful.

3) It would be useful to confirm that both chronic OT neuron depletion as well as chemogenetic activation indeed affect the activity of OT neurons by assessing a functional parameter, i.e. OT staining, or plasma OT levels. Although chemogenetic activation of PVN OT neurons has been shown to elevate peripheral and central OT concentrations (Grund et al., 2019), is this still the case after repeated acute activation over 3 weeks? To which degree are OT neurons depleted by then?

4) In this context, the authors also describe that "celastrol may regulate the performance of certain ion channels, thus enhancing Oxt neuron firing in response to physiological stimuli". In the context of their study, what is the physiological stimulus? Does celastrol activate also baseline neuronal activity? Does celastrol also trigger OT secretion in vivo? Here, answers to these questions need to be given.

5) Please provide evidence that celastrol selectively affects OT neurons in the PVN and not any other neurons in the brain, as it was administered icv, and not only into the PVN

*Reviewer #1:*

The authors were hoping to be able to demonstrate that oxytocin neurons in the paraventricular nucleus can impact anxiety and modulate colitis-associated cancer (CAC) progression using a mouse model. They also identified a compound, celastrol, that can activate oxytocin neurons and reduce CAC tumor progression through an oxytocin-like pathway. Lastly, they were hoping to be able to show that celastrol can suppress tumor activity by inhibiting sympathetic nervous system activation.

The major strengths to this body of work lie in the novelty of the hypotheses being examined and the number of sophisticated approaches taken to test their hypotheses. The investigators were largely able to achieve their goals given that they identified previous findings showing that impairments in oxytocin signaling can increase anxiety using a number of behavioral approaches. They were able to demonstrate an increase in CAC progression using a mouse model with depleted oxytocin signaling within the PVN. They then found that stimulation of PVN oxytocin neurons can inhibit CAC progression by suppressing cell proliferation and promoting cell apoptosis. They found that treatment with celastrol could excite PVN oxytocin neurons and that brain treatment with celastrol can suppress CAC progression in mice. In addition, they found that hypothalamic oxytocin neurons are required in the anti-tumor effect of celastrol.

The main weakness I identified has to do with their last question as to whether the SNS contributes to these effects. I think that the authors could have more fully identified that the neurons in question were actually sympathetic neurons using a marker such as tyrosine hydroxylase. In addition, I think they could have lesioned SNS innervation to the colon through a surgical or chemical (6-OHDA) to more fully determine the impact of defective SNS innervation in their model.

I think that the authors achieved the majority of their aims in question with the exception of their last aim addressing the role of the sympathetic nervous system.

This body of work will be largely impactful to gastrointestinal physiologists, cancer biologists and neuroscientists as it has high clinical relevance given the therapeutic potential of oxytocin treatment. It reveals a potential novel treatment for colitis-associated cancer and the mechanisms that may contribute to these effects. The authors used a series of compelling experimental manipulations (genetic, chemogenetic, pharmacological and electrophysiological) and the utility of these approaches in this particular context could be very useful to the scientific community. They used an elegant number of approaches as a way to dissect the circuit and to largely support the key claims of the paper.

The authors have provided an impressive body of work to dissect the relevance of oxytocin neurons in modulating colitis-associated cancer and the extent to which a compound found to reduce CAC progression works through the oxytocin pathway. The authors should be commended for their thorough examination using a variety of genetic, chemogenetic, pharmacological and electrophysiological as a way to dissect the circuit and to largely support the key claims of the paper.

I think that the main concerns I have pertain to the role of the sympathetic nervous system suppression in contributing to these effects. I think the authors could first identify that sympathetic neurons were impacted using a marker specific to catecholamine neurons (tyrosine hydroxylase, for example). Secondly, I think they could assess the effects in animals that lack SNS innervation to the target tissues in question using either surgical or chemical ablation (6-OHDA) approaches as opposed to a β-2 receptor agonist.

*Reviewer #2:*

Pan et al., studied the consequences of manipulations of hypothalamic oxytocin (OT) neurons on pharmacologically induced colorectal cancer progression in mice. They use genetic and chemogenetic approaches to either chronically deplete all brain OT neurons or to selectively activate PVN OT neurons. Further, treatment with celastrol, a pentacyclic triterpenoid, which should excite OT neurons and which they applied into the cerebral ventricles daily over 3 weeks also inhibited colorectal cancer progression, an effect, which was attenuated in OT neuron-depleted mice.

Furthermore, brain treatment with celastrol suppresses neuronal activity in the celiac-superior mesenteric ganglion, and activation of β2 adrenergic receptor abolished the anti-tumor effect of centrally administered celastrol. In sum, the authors intend to show, by manipulation of the OT system its contribution to colorectal cancer progression. Although the experiments give some potential insights into the role of OT neurons in the immune responsiveness and the progression of colorectal cancer, novel causal relationships between the different findings are rather missing.

In order to increase the impact of the paper, especially to show causalities between OT manipulations and progression in colorectal cancer, the following points might be taken into consideration:

– What is missing is a proposed causal mechanism of the anticancer effect of OT neuron activation. Is it the attenuation of the activity of the HPA axis, as repeatedly shown by OT, is it the reduction in chronic stress levels mediated by OT, are anti-inflammatory effects involved or other effects on the immune system ? An experiment blocking OT receptors (centrally or within selected brain regions) or an experiment manipulating corticosterone levels during OT neuronal activation or depletion might be helpful.

– It would be useful to confirm that both chronic OT neuron depletion as well as chemogenetic activation indeed affect the activity of OT neurons by assessing a functional parameter, i.e. OT staining, or plasma OT levels. Although chemogenetic activation of PVN OT neurons has been shown to elevate peripheral and central OT concentrations (Grund et al., 2019), is this still the case after repeated acute activation over 3 weeks? To which degree are OT neurons depleted by then?

– In this context, the authors also describe that "celastrol may regulate the performance of certain ion channels, thus enhancing Oxt neuron firing in response to physiological stimuli". In the context of their study, what is the physiological stimulus? Does celastrol activate also baseline neuronal activity? Does celastrol also trigger OT secretion in vivo? Here, answers to these questions need to be given.

– Please provide evidence that celastrol selectively affects OT neurons in the PVN and not any other neurons in the brain, as it was administered icv, and not only into the PVN.

– What is the evidence that celastrol-induced suppresses of the activity of sympathetic neurons in the CG-SMG ganglion is mediated by OT? Inhibition or depletion of OT neurons may affect many other systems of the brain, such as the CRF system, which may result in elevated stress levels.

– In addition to negative mood, also other factors, which are significantly regulated by OT, need to be considered such as social support and chronic stress. In fact, chronic stress in mice was repeatedly described to induce colitis and to enhance colorectal cancer by the Reber group.In contrast, social support, mediated by OT, was shown to attenuate cancerogenesis and stress responses (Heinrichs et al.,).These aspects might be thoroughly considered and discussed.

– The link to negative moods, including depression and stress, repeatedly described in the introduction and discussion, remains vague, as mice were not manipulated to induce a state of increased anxiety or chronic stress.

In order to increase the impact of the paper, and to justify the conclusions drawn, especially to show causalities between OT manipulations and progression in colorectal cancer, the following points might be taken into consideration:

– What is missing is a proposed causal mechanism of the anticancer effect of OT neuron activation. Is it the attenuation of the activity of the HPA axis, as repeatedly shown by OT, is it the reduction in chronic stress levels mediated by OT, are anti-inflammatory effects involved or other effects on the immune system ? An experiment blocking OT receptors (centrally or within selected brain regions) or an experiment manipulating corticosterone levels during OT neuronal activation or depletion might be helpful.

– It would be useful to confirm that both chronic OT neuron depletion as well as chemogenetic activation indeed affect the activity of OT neurons by assessing a functional parameter, i.e. OT staining, or plasma OT levels. Although chemogenetic activation of PVN OT neurons has been shown to elevate peripheral and central OT concentrations (Grund et al., 2019), is this still the case after repeated acute activation over 3 weeks? To which degree are OT neurons depleted by then?

– In this context, the authors also describe that "celastrol may regulate the performance of certain ion channels, thus enhancing Oxt neuron firing in response to physiological stimuli". In the context of their study, what is the physiological stimulus? Does celastrol activate also baseline neuronal activity? Does celastrol also trigger OT secretion in vivo? Here, answers to these questions need to be given.

– Please provide evidence that celastrol selectively affects OT neurons in the PVN and not any other neurons in the brain, as it was administered icv, and not only into the PVN.

– What is the evidence that celastrol-induced suppresses of the activity of sympathetic neurons in the CG-SMG ganglion is mediated by OT? Inhibition or depletion of OT neurons may affect many other systems of the brain, such as the CRF system, which may result in elevated stress levels.

– In addition to negative mood, also other factors, which are significantly regulated by OT, need to be considered such as social support and chronic stress. In fact, chronic stress in mice was repeatedly described to induce colitis and to enhance colorectal cancer by the Reber group.In contrast, social support, mediated by OT, was shown to attenuate cancerogenesis and stress responses (Heinrichs et al.,).These aspects might be thoroughly considered and discussed.

*Reviewer #3:*

The authors note the role of anxiety in cancer risk and hypothesize that this role might be mediated to some extent via oxytocin neurons. To examine this hypothesis the authors attempted to examine the extent to which oxytocin neurons might modulate incidence and progression of colitis induced cancer. To answer this question they looked at effects of both positive and negative manipulation of oxytocin neurons. They observed that inhibition enabled cancer progression and further that activation prevented cancer progression.

Initially the authors demonstrate that genetically enabled lesioning of oxytocin neurons allows increases colitis associated cancer progression. Then they further demonstrate that chemogenetic activation of oxytocin neurons decreases colitis associated cancer progression. To validate a novel reagent, they then demonstrate that a novel herbal isolate activates oxytocin neurons and also decreases colitis associated cancer progression. They demonstrate that lesioning of oxytocin neurons abrogates this effect. Finally, they demonstrated that their novel compound inhibited SNS outflow and that bypass of this inhibition with the β-adrenergic agonist abrogated its prevention of colitis associated cancer.

Strengths of the work demonstrating include multiple manipulations of oxytocin neuron activity on colitis associated cancer. One relatively weakness of the work is that the sympathetic nervous system effect was only examined in the context of their novel reagent.

This work provides a basis for how anxiety might alter cancer risk.

Overall this is a strong manuscript. As noted above, one weakness is the demonstration of oxytocin neuron downstream effects on the SNS and bypass by the β-adrenergic agonist only using the novel herbal reagent (rather than, for instance, in the DREADD-dependent model).

[Editors' note: further revisions were suggested prior to acceptance, as described below.]

Thank you for resubmitting your work entitled "Stimulation of hypothalamic oxytocin neurons suppresses colorectal cancer progression in mice" for further consideration by eLife. Your revised article has been evaluated by Mone Zaidi (Senior Editor) and a Reviewing Editor.

The manuscript has addressed the responses of the previous reviewers and has been substantially improved. However, there is one remaining issue that needs to be addressed, as outlined below:

The authors should remove Figure 3S1-A-B as it does not provide helpful information to the paper. I have a concern over the use of TH intensity as a way to measure SNS activity in IBAT. As indicated in the paper by Vaughan and Bartness (Methods Enzymol, 537: 199-235, 2014): "NETO is used as a direct neurochemical measure of sympathetic drive; as noted above, there is no surrogate for this method of assessment except for direct measures of sympathetic nerve activity electrophysiologically". The authors should remove mention of SNS activity within IBAT unless they can provide this assessment via NETO or electrophysiology. As mentioned earlier, the information provided is not the currently accepted approach to assess SNS in animals with IBAT denervation.

---

## [Author Response]

Essential revisions:In order to increase the impact of the paper, and to justify the conclusions drawn, especially to show causalilties between OT manipulations and progression in colorectal cancer, the following points should be taken into consideration:1) The role of the sympathetic nervous system suppression in contributing to these effects is not yet clear. The authors could first identify that sympathetic neurons were impacted using a marker specific to catecholamine neurons (tyrosine hydroxylase, for example). Secondly, the authors could assess the effects in animals that lack SNS innervation to the target tissues in question using either surgical or chemical ablation (6-OHDA) approaches as opposed to a β-2 receptor agonist. Related to this is that the sympathetic nervous system effect was only examined in the context of their novel reagent (rather than, for instance, in the DREADD-dependent model).

We gratefully thank the reviewers for these comments.

1) During this revision, we have assessed the effect of excitation of Oxt^PVN^ neurons, or transection of preganglionic fiber of CG-SMG on the activities of tyrosine hydroxylase (TH)-positive neurons in the CG-SMG. The data indicate that excitation of Oxt^PVN^ neurons rapidly suppressed the activities of TH-positive neurons in the CG-SMG (Figure 3A,B). In a separate experiment, we showed that i.c.v. administration of celastrol readily suppressed the activities of TH-positive neurons in the CG-SMG, and transection of the preganglionic fiber could significantly attenuate this effect (Figure 7—figure supplement 1A-C).

2) To address the reviewer’s second question, we elected to surgically remove CG-SMG in the *Oxt^Cre^* and *Oxt^Cre^;DTA* mice (Figure 3C-K). In agreement with our early observation, depletion of Oxt neurons promoted colitis-associated cancer (CAC) development in mice (Figure 3E-K). After the resection of CG-SMG, this effect was markedly abrogated (Figure 3E-K).

3) Following the reviewer’s suggestion, we examined the relationship between Oxt^PVN^ neurons and β2-adrenergic receptor (β2AR) in the progression of CAC. The data indicate that the DREADD-mediated activation of Oxt^PVN^ neurons (Figure 6A,B) suppressed CAC progression in mice (Figure 6C-J). Notably, i.p. administration of isoprenaline, a β2AR agonist, could significantly attenuate this effect (Figure 6C-J). These data suggest that suppression of β2AR activity is crucial for Oxt^PVN^ neuron activation to restrict CAC progression.

2) What is missing is a proposed causal mechanism of the anticancer effect of OT neuron activation. Is it the attenuation of the activity of the HPA axis, as repeatedly shown by OT, is it the reduction in chronic stress levels mediated by OT, are anti-inflammatory effects involved or other effects on the immune system ? An experiment blocking OT receptors (centrally or within selected brain regions) or an experiment manipulating corticosterone levels during OT neuronal activation or depletion might be helpful.

We gratefully thank reviewer #2 for these very helpful comments.

1) In this revision, we have assessed the activity of the HPA axis. Our data indicate that depletion of Oxt neurons resulted in the elevation of circulating ACTH and corticosterone levels in mice (Figure 1—figure supplement 1L,M). Conversely, chemogenetic approach-mediated excitation of Oxt^PVN^ neurons could lead to a significant decrease of ACTH and corticosterone levels in systemic circulation (Figure 1—figure supplement 2K,L). These data suggest that the HPA axis may play a role in the modulation of tumor progression by Oxt^PVN^ neurons.

2) In agreement with the changes in the HPA axis, our assessments show that mice deficient for Oxt neurons exhibited an elevated anxiety level (Figure 1—figure supplement 1A-C), while excitation of Oxt^PVN^ neurons in *Oxt^Cre^* mice had an anxiolytic effect (Figure 1—figure supplement 2A-C).

3) Also, the *Oxt^Cre^* mice were injected with control or hM3Dq AAV into the PVN, and then were i.p. administered with CNO every other day for 3 consecutive weeks (Figure 1—figure supplement 2D). The tumor tissues were then harvested and immune cells were assessed. The data show that the number of CD8^+^ T cells was remarkably increased in the tumor tissue of the mice with Oxt^PVN^ neuron activation, whereas other types of immune cell were not significantly impacted (Figure 1—figure supplement 3). These data suggest that excitation of Oxt^PVN^ neurons in the brain may bestow its beneficial effect by promoting the anti-tumor immunity.

4) Following the reviewer’s suggestion, we carried out an experiment in which L-368,899, a selective Oxt receptor (OTR) antagonist, was used to block OTR in the mouse brain (Figure 2 and Figure 2—figure supplement 1). The data show that CAC progression was inhibited in the *Oxt^Cre^* mice in which Oxt^PVN^ neurons had been stimulated (Figure 2C-J). Notably, blockade of OTR in the brain, which was achieved by injecting L-368,899 into the third ventricle, could markedly abolish the tumor suppression effect of Oxt^PVN^ neuron activation (Figure 2C-J). These data indicate that brain OTR is crucial for activation of Oxt^PVN^ neurons to suppress CAC progression in mice.

3) It would be useful to confirm that both chronic OT neuron depletion as well as chemogenetic activation indeed affect the activity of OT neurons by assessing a functional parameter, i.e. OT staining, or plasma OT levels. Although chemogenetic activation of PVN OT neurons has been shown to elevate peripheral and central OT concentrations (Grund et al., 2019), is this still the case after repeated acute activation over 3 weeks? To which degree are OT neurons depleted by then?

We thank reviewer #2 for these valid points. To address them, we have carried out both immunofluorescent staining and Oxt EIA assays. (1) Regarding Oxt neuron depletion, the immunofluorescent staining data demonstrate that, at the end of the experiment, ⁓94% of the Oxt neurons had been lesioned in the PVN of the *Oxt^Cre^;DTA* mice (Figure 1B,C), in which plasma Oxt was barely detectable (Figure 1—figure supplement 1G). (2) With regard to the chemogenetic activation of Oxt^PVN^ neurons, after a 3-week treatment of CNO, the majority of Oxt^PVN^ neurons were excited (Figure 1I,J), and plasma Oxt level was elevated in the hM3Dq AAV-injected mice (Figure 1—figure supplement 2E). Together, these data indicate that the employed experimental models could work as expected. We also cited the study by Grund and colleagues in the revised manuscript.

4) In this context, the authors also describe that "celastrol may regulate the performance of certain ion channels, thus enhancing Oxt neuron firing in response to physiological stimuli". In the context of their study, what is the physiological stimulus? Does celastrol activate also baseline neuronal activity? Does celastrol also trigger OT secretion in vivo? Here, answers to these questions need to be given.

In the slice electrophysiology experiments, current injection ranging from 20 to 200 pA was used to test the excitability of Oxt^PVN^ neurons. Previous work indicated that physiological stimuli, such as social touch^1^, tactile stimuli^2^, feeding^3^ and leptin^4^ could lead to the excitation of Oxt neurons. Here, the electrical stimuli were utilized to mimic the excitatory inputs in response to natural stimuli mentioned above. Our results suggest that celastrol could elevate the responsiveness to the same stimuli. We apologize for not having described this clearly. Our data indicate that i.c.v. administration of celastrol could excite Oxt^PVN^ neurons (percentage of c-Fos-positive Oxt^PVN^ neurons of total Oxt^PVN^ neurons: vehicle, 12.3±2.0%; celastrol, 34.7±6.6%. *P*=0.01, n=5 mice per group), suggesting that it can activate these neurons.

To assess the effect of celastrol on Oxt secretion, we chose to use an ex vivo Oxt release assay, since this method has been established in our laboratory^5^. The PVN slices were dissected from adult male C57 BL/6 mice, and then were balanced in normal Locke’s solution. Thereafter, the tissues were incubated in the same solution supplemented with celastrol. The contents of Oxt in these solutions were then determined using an Oxt EIA kit. Indeed, treatment with celastrol could enhance Oxt secretion from the PVN slices (Figure 4—figure supplement 1F).

5) Please provide evidence that celastrol selectively affects OT neurons in the PVN and not any other neurons in the brain, as it was administered icv, and not only into the PVN

We thank reviewer #2 for this valid suggestion. To address this question, adult male C57 BL/6 mice were i.c.v. administered with celastrol versus vehicle control. Two hours later, mice were perfused with 4% paraformaldehyde, and then brain tissues were sectioned. Immunofluorescent staining for c-Fos demonstrates that treatment with celastrol triggered excitation of neurons in the PVN, but not other hypothalamic nuclei (Figure 4—figure supplement 1A,B). In combination with our electrophysiological data (Figure 4A-G), these results suggest that celastrol could selectively regulate the activities of Oxt neurons in the PVN.

Reviewer #1:This body of work will be largely impactful to gastrointestinal physiologists, cancer biologists and neuroscientists as it has high clinical relevance given the therapeutic potential of oxytocin treatment. It reveals a potential novel treatment for colitis-associated cancer and the mechanisms that may contribute to these effects. The authors used a series of compelling experimental manipulations (genetic, chemogenetic, pharmacological and electrophysiological) and the utility of these approaches in this particular context could be very useful to the scientific community. They used an elegant number of approaches as a way to dissect the circuit and to largely support the key claims of the paper.

We gratefully thank reviewer #1 for these encouraging comments.

The authors have provided an impressive body of work to dissect the relevance of oxytocin neurons in modulating colitis-associated cancer and the extent to which a compound found to reduce CAC progression works through the oxytocin pathway. The authors should be commended for their thorough examination using a variety of genetic, chemogenetic, pharmacological and electrophysiological as a way to dissect the circuit and to largely support the key claims of the paper.

We are very grateful to reviewer #1 for these positive comments.

I think that the main concerns I have pertain to the role of the sympathetic nervous system suppression in contributing to these effects. I think the authors could first identify that sympathetic neurons were impacted using a marker specific to catecholamine neurons (tyrosine hydroxylase, for example). Secondly, I think they could assess the effects in animals that lack SNS innervation to the target tissues in question using either surgical or chemical ablation (6-OHDA) approaches as opposed to a β-2 receptor agonist.

We appreciate reviewer #1 for these valid suggestions.

1) During this revision, we have assessed the effect of the excitation of Oxt^PVN^ neurons, or the transection of preganglionic fiber of CG-SMG on the activities of tyrosine hydroxylase (TH)-positive neurons in the CG-SMG. The data indicate that excitation of Oxt^PVN^ neurons rapidly suppressed the activities of TH-positive neurons in the CG-SMG (Figure 3A,B). In a separate experiment, we showed that i.c.v. administration of celastrol readily suppressed the activities of TH-positive neurons in the CG-SMG, and the transection of the preganglionic fiber could significantly attenuate this effect (Figure 7—figure supplement 1A-C).

2) To address the reviewer’s second question, we elected to surgically remove CG-SMG in the *Oxt^Cre^* and *Oxt^Cre^;DTA* mice (Figure 3C-K). In agreement with our early observation, depletion of Oxt neurons promoted colitis-associated cancer (CAC) development in mice (Figure 3E-K). After the resection of CG-SMG, this effect was significantly abrogated (Figure 3E-K).

Reviewer #2:In order to increase the impact of the paper, and to justify the conclusions drawn, especially to show causalilties between OT manipulations and progression in colorectal cancer, the following points might be taken into consideration:– What is missing is a proposed causal mechanism of the anticancer effect of OT neuron activation. Is it the attenuation of the activity of the HPA axis, as repeatedly shown by OT, is it the reduction in chronic stress levels mediated by OT, are anti-inflammatory effects involved or other effects on the immune system? An experiment blocking OT receptors (centrally or within selected brain regions) or an experiment manipulating corticosterone levels during OT neuronal activation or depletion might be helpful.

We gratefully thank reviewer #2 for these very helpful comments.

1) In this revision, we have assessed the activity of the HPA axis. Our data indicate that depletion of Oxt neurons resulted in the elevation of the circulating ACTH and corticosterone levels in mice (Figure 1—figure supplement 1L,M). Conversely, chemogenetic approach-mediated excitation of Oxt^PVN^ neurons could significantly decrease ACTH and corticosterone levels in systemic circulation (Figure 1—figure supplement 2K,L). These data suggest that the HPA axis may play a role in the modulation of tumor progression by Oxt^PVN^ neurons.

2) In agreement with the changes in the HPA axis, our assessments show that mice deficient for Oxt neuron exhibited an elevated anxiety level (Figure 1—figure supplement 1A-C), while excitation of Oxt^PVN^ neurons in *Oxt^Cre^* mice had an anxiolytic effect (Figure 1—figure supplement 2A-C).

3) Also, the *Oxt^Cre^* mice were injected with control or hM3Dq AAV into the PVN, and then were i.p. administered with CNO every other day for 3 consecutive weeks (Figure 1—figure supplement 2D). The tumor tissues were then harvested and immune cells were assessed. The data show that the number of CD8^+^ T cells was remarkably increased in the tumor tissue of the mice with Oxt^PVN^ neuron activation, whereas other types of immune cell were not significantly impacted (Figure 1—figure supplement 3). These data suggest that excitation of Oxt^PVN^ neurons in the brain may bestow its beneficial effect by promoting anti-tumor immunity.

4) Following the reviewer’s suggestion, we carried out an experiment in which L-368,899, a selective Oxt receptor (OTR) antagonist, was used to block OTR in the mouse brain (Figure 2 and Figure 2—figure supplement 1). The data show that CAC progression was inhibited in the *Oxt^Cre^* mice in which Oxt^PVN^ neurons had been stimulated (Figure 2C-J). Notably, blockade of OTR in the brain, which was achieved by injecting L-368,899 into the third ventricle, could markedly abolish the tumor suppression effect of Oxt^PVN^ neuron activation (Figure 2C-J). These data indicate that brain OTR is crucial for activation of Oxt^PVN^ neurons to suppress CAC progression in mice.

– It would be useful to confirm that both chronic OT neuron depletion as well as chemogenetic activation indeed affect the activity of OT neurons by assessing a functional parameter, i.e. OT staining, or plasma OT levels. Although chemogenetic activation of PVN OT neurons has been shown to elevate peripheral and central OT concentrations (Grund et al., 2019), is this still the case after repeated acute activation over 3 weeks? To which degree are OT neurons depleted by then?

We thank reviewer #2 for these valid points. To address them, we have carried out both immunofluorescent staining and Oxt EIA assays. (1) Regarding Oxt neuron depletion, the immunofluorescent staining data demonstrate that, at the end of the experiment, ⁓94% of the Oxt neurons had been lesioned in the PVN of the *Oxt^Cre^;DTA* mice (Figure 1B,C), in which plasma Oxt was barely detectable (Figure 1—figure supplement 1G). (2) With regard to the chemogenetic activation of Oxt^PVN^ neurons, after a 3-week treatment of CNO, the majority of Oxt^PVN^ neurons were excited (Figure 1I,J), and plasma Oxt level was elevated in the hM3Dq AAV-injected mice (Figure 1—figure supplement 2E). Together, these data indicate that the employed experimental models could work as expected. We also cited the study by Grund and colleagues in the revised manuscript.

– In this context, the authors also describe that "celastrol may regulate the performance of certain ion channels, thus enhancing Oxt neuron firing in response to physiological stimuli". In the context of their study, what is the physiological stimulus? Does celastrol activate also baseline neuronal activity? Does celastrol also trigger OT secretion in vivo? Here, answers to these questions need to be given.

In the slice electrophysiology experiments, current injection ranging from 20 to 200 pA was used to test the excitability of Oxt^PVN^ neurons. Previous work indicated that physiological stimuli, such as social touch^1^, tactile stimuli^2^, feeding^3^ and leptin^4^ could lead to the excitation of Oxt neurons. Here, the electrical stimuli were utilized to mimic the excitatory inputs in response to natural stimuli mentioned above. Our results suggest that celastrol could elevate the responsiveness to the same stimuli. We apologize for not having described this clearly. Our data indicate that i.c.v. administration of celastrol could excite Oxt^PVN^ neurons (percentage of c-Fos-positive Oxt^PVN^ neurons of total Oxt^PVN^ neurons: vehicle, 12.3±2.0%; celastrol, 34.7±6.6%. *P*=0.01, n=5 mice per group).

To assess the effect of celastrol on Oxt secretion, we chose to use an ex vivo Oxt release assay, since this method has been established in our laboratory^5^. The PVN slices were dissected from adult male C57 BL/6 mice, and then were balanced in normal Locke’s solution. Thereafter, the tissues were incubated in the same solution supplemented with celastrol. The contents of Oxt in these solutions were then determined using an Oxt EIA kit. Indeed, treatment with celastrol could enhance Oxt secretion from the PVN slices (Figure 4—figure supplement 1F).

– Please provide evidence that celastrol selectively affects OT neurons in the PVN and not any other neurons in the brain, as it was administered icv, and not only into the PVN.

We thank reviewer #2 for this valid suggestion. To address this question, adult male C57 BL/6 mice were i.c.v. administered with celastrol versus vehicle control. Two hours later, mice were perfused with 4% paraformaldehyde, and then brain tissues were sectioned. Immunofluorescent staining for c-Fos demonstrates that treatment with celastrol triggered excitation of neurons in the PVN, but not other hypothalamic nuclei (Figure 4—figure supplement 1A,B). In combination with our electrophysiological data (Figure 4A-G), these results suggest that celastrol could selectively regulate the activities of Oxt neurons in the PVN.

– What is the evidence that celastrol-induced suppression of the activity of sympathetic neurons in the CG-SMG ganglion is mediated by OT? Inhibition or depletion of OT neurons may affect many other systems of the brain, such as the CRF system, which may result in elevated stress levels.

1) To address the reviewer’s point, male adult C57 BL/6 mice were implanted with a guide cannula directed to third ventricle. After recovery, these mice were i.c.v. injected with aCSF or L-368,899, the OTR antagonist. An hour later, the 6-min control spiking activities were acquired from neurons of the CG-SMG, before celastrol application through the pre-implanted guide cannula. This in vivo single-unit recordings data demonstrates that i.c.v. administration of celastrol decreased the firing frequency of neurons in the CG-SMG, and that pre-treatment with OTR antagonist could significantly attenuate this effect (Figure 7D-G). These results indicate that brain Oxt is important for celastrol to regulate the neuronal activity in the CG-SMG.

2) During this revision, we measured the plasma ACTH and corticosterone levels in the *Oxt^Cre^* and *Oxt^Cre^;DTA* mice. The data display that both ACTH and corticosterone levels were elevated in the *Oxt^Cre^;DTA* mice compared to their levels in the controls (Figure 1—figure supplement 1L,M). This increased activity of the HPA axis may lead to elevated stress level, and then contribute to the development of CAC. However, this needs further investigations.

– In addition to negative mood, also other factors, which are significantly regulated by OT, need to be considered such as social support and chronic stress. In fact, chronic stress in mice was repeatedly described to induce colitis and to enhance colorectal cancer by the Reber group.In contrast, social support, mediated by OT, was shown to attenuate cancerogenesis and stress responses (Heinrichs et al.,).These aspects might be thoroughly considered and discussed.

Following the reviewer’s suggestion, we have included the discussions of the effects of chronic stress, especially those studies by the Reber group, and social support (by Heinrichs et al.,) on colitis and colorectal cancer in the revised manuscript.

Reviewer #3:The authors note the role of anxiety in cancer risk and hypothesize that this role might be mediated to some extent via oxytocin neurons. To examine this hypothesis the authors attempted to examine the extent to which oxytocin neurons might modulate incidence and progression of colitis induced cancer. To answer this question they looked at effects of both positive and negative manipulation of oxytocin neurons. They observed that inhibition enabled cancer progression and further that activation prevented cancer progression.Initially the authors demonstrate that geneticly enabled lesioning of oxytocin neurons allows increases colitis associated cancer progression. Then they further demonstrate that chemogenetic activation of oxytocin neurons decreases colitis associated cancer progression. To validate a novel reagent, they then demonstrate that a novel herbal isolate activates oxytocin neurons and also decreases colitis associated cancer progression. They demonstrate that lesioning of oxytocin neurons abrogates this effect. Finally, they demonstrated that their novel compound inhibited SNS outflow and that bypass of this inhibition with the β-adrenergic agonist abrogated its prevention of colitis associated cancer.Strengths of the work demonstrating include multiple manipulations of oxytocin neuron activity on colicitis associated cancer. One relatively weakness of the work is that the sympathetic nervous system effect was only examined in the context of their novel reagent.This work provides a basis for how anxiety might alter cancer risk.

Overall this is a strong manuscript. As noted above, one weakness is the demonstration of oxytocin neuron downstream effects on the SNS and bypass by the β-adrenergic agonist only using the novel herbal reagent (rather than, for instance, in the DREADD-dependent model).

We thank reviewer #3 for this comment. Following the reviewer’s suggestion, we examined the relationship between Oxt neurons in the PVN and β2-adrenergic receptor (β2AR) in the progression of colitis-associated cancer (CAC). The data indicate that the DREADD-mediated activation of Oxt neurons in the PVN (Figure 6A,B) suppressed CAC progression in mice (Figure 6C-J). Notably, i.p. administration of isoprenaline, a β2AR agonist, could significantly attenuate this effect (Figure 6C-J). These data suggest that suppression of β2AR activity is crucial for activation of Oxt neurons in the PVN to restrict CAC progression in mice.

References

1. Tang, Y.*, et al.,* Social touch promotes interfemale communication via activation of parvocellular oxytocin neurons. *Nat Neurosci* 23, 1125-1137 (2020).

2. Okabe, S., Yoshida, M., Takayanagi, Y. and Onaka, T. Activation of hypothalamic oxytocin neurons following tactile stimuli in rats. *Neurosci Lett* 600, 22-27 (2015).

3. Johnstone, L.E., Fong, T.M. and Leng, G. Neuronal activation in the hypothalamus and brainstem during feeding in rats. *Cell Metab* 4, 313-321 (2006).

4. Blevins, J.E., Schwartz, M.W. and Baskin, D.G. Evidence that paraventricular nucleus oxytocin neurons link hypothalamic leptin action to caudal brain stem nuclei controlling meal size. *Am J Physiol Regul Integr Comp Physiol* 287, R87-96 (2004).

5. Wu, L.*, et al.,* Caffeine inhibits hypothalamic A_1_R to excite oxytocin neuron and ameliorate dietary obesity in mice. *Nat Commun* 8, 15904 (2017).

[Editors' note: further revisions were suggested prior to acceptance, as described below.]

The authors should remove Figure 3S1-A-B as it does not provide helpful information to the paper. I have a concern over the use of TH intensity as a way to measure SNS activity in IBAT. As indicated in the paper by Vaughan and Bartness (Methods Enzymol, 537: 199-235, 2014): "NETO is used as a direct neurochemical measure of sympathetic drive; as noted above, there is no surrogate for this method of assessment except for direct measures of sympathetic nerve activity electrophysiologically". The authors should remove mention of SNS activity within IBAT unless they can provide this assessment via NETO or electrophysiology. As mentioned earlier, the information provided is not the currently accepted approach to assess SNS in animals with IBAT denervation.

We gratefully thank the editors for these valid comments. Following the suggestion, we have deleted Figure 3-figure supplement 1A,B in the previous manuscript. We have also edited the main text and other contents (eg., the Methods section and the figure legend for Figure 3-figure supplement 1) to reflect this change.